# Approximate Message Passing for Bayesian Neural Networks

## Abstract

Bayesian methods for learning predictive models have the ability to consider both sources of uncertainty (i.e., data and model uncertainty) within a single framework and thereby provide a powerful tool for decision-making. Bayesian neural networks (BNNs) hold great potential for training data efficiency due to full uncertainty quantification, making them promising candidates for more data-efficient AI in data-constrained settings such as reinforcement learning in the physical world. However, current computational approaches for learning BNNs often face limitations such as overconfidence, sensitivity to hyperparameters, and posterior collapse, highlighting the need for alternative computational approaches. In this paper, we introduce a novel method that leverages approximate message passing (MP) of a full factorized neural network model using mixed approximations to overcome these problems while maintaining data efficiency. Our framework supports convolutional neural networks while addressing the issue of double-counting training data, which has been a key source of overconfidence in prior work. We demonstrate the data-efficiency of our method on multiple benchmark datasets in comparison to state-of-the-art methods for learning neural networks.

## 1 Introduction

Deep learning models have achieved impressive results across various domains, including natural language processing (Vaswani et al., 2023), computer vision (Ravi et al., 2024), and autonomous systems (Bojarski et al., 2016). Yet, they often produce overconfident but incorrect predictions, particularly in ambiguous or out-of-distribution scenarios. Without the ability to effectively quantify uncertainty, this can foster both over-reliance and under-reliance on models, as users stop trusting their outputs altogether(Zhang et al., 2024), and in high-stakes domains like healthcare or autonomous driving, its application can be dangerous (Henne et al., 2020). To ensure safer deployment in these settings, models must not only predict outcomes, but also express how uncertain they are about those predictions to allow for informed decision-making.

Bayesian neural networks (BNNs) offer a principled way to quantify both sources of uncertainty (i.e., data and model uncertainty) by capturing a posterior distribution over the model's weights, rather than relying on point estimates of all weights as in traditional neural network training. This allows BNNs to express epistemic uncertainty, the model's lack of knowledge about the underlying data distribution. Current methods for posterior approximation largely fall into two categories: sampling-based methods, such as Hamiltonian Monte Carlo (HMC), and deterministic approximation methods such as variational inference (VI). While sampling methods are usually computationally expensive, VI has become increasingly scalable (Shen et al., 2024). However, VI is not without limitations: It often struggles with overconfidence (Papamarkou et al., 2024), and it can struggle to break symmetry when multiple modes are close (Zhang et al., 2018). Mean-field approaches, commonly used in VI, are prone to posterior collapse (Kurle et al., 2022; Coker et al., 2022). Additionally, VI often requires complex hyperparameter tuning (Osawa et al., 2019), which complicates its practical deployment in real-world settings. These challenges motivate the need for alternative approaches that can address shortcomings of VI while maintaining its scalability.

In contrast, expectation propagation (EP) (Minka, 2001) is an approximate probabilistic inference technique that suffers less from these problems. In order to scale EP, it is best cast as a local approximation method for message passing (MP) in a factor graph model (Kschischang et al., 2001)

of a neural network with weights $\boldsymbol{w}_k$. A factor graph models the joint density $p(\boldsymbol{w}_1, \ldots, \boldsymbol{w}_n)$ of all weight vectors into a product of functions $f_j$ on subsets of random weight variables $W_1, \ldots, W_n$. Thus, a factor graph is a bipartite graph where each factor $f_j$ is connected with the variables they depend on. The following recursive equations yield a computationally efficient algorithm to compute all marginals $p_W(\cdot)$ for acyclic factor graphs:

$$p_W(\boldsymbol{w}) = \prod\nolimits_{f \in N_W} m_{f \to W}(\boldsymbol{w})$$

$$m_{W \to f}(\boldsymbol{w}) = \int f(\boldsymbol{w}, \boldsymbol{w}'_1, \ldots, \boldsymbol{w}'_k) \left[ \prod\nolimits_{W' \in N_f \setminus \{W\}} m_{W' \to f}(\boldsymbol{w}') \right] d\boldsymbol{w} \, d\boldsymbol{w}'_1 \cdots d\boldsymbol{w}'_k \,,$$

where $m_{W' \to f}(\boldsymbol{w}') = \prod_{g \in N_V \setminus \{f\}} m_{g \to V}(\boldsymbol{w}')$ and $N_v$ denotes the neighborhood of node $v$ (i.e., either a variable $W$ or a factor $f$). Since exact messages $m_{f \to W}(\cdot)$ are often intractable and factor graphs are rarely acyclic, the recursive application of the above computations are not applicable. Instead, messages $m_{f \to W}(\cdot)$ and marginals $p_W(\cdot)$ are typically approximated by some family of distributions that has few parameters (e.g., Gaussians) and where close approximations to this integral can be efficiently computed. However, applying MP in practice presents two main challenges for practitioners: the need to derive (approximate) message equations for $m_{f \to W}(\cdot)$ and the complexity of implementing MP compared to other methods.

**Contributions**   Our contributions can be summarized as follows:

1. We propose a novel message-passing framework for BNNs and derive computationally efficient message equations for all necessary factors.
2. We implement our method in Julia for both CPU and GPU, and demonstrate its general applicability to convolutional neural networks (CNNs) and multilayer perceptrons (MLPs) while avoiding the double-counting problem.
3. We find that that our method is overall competitive with the state-of-the-art baselines AdamW and IVON as well as Deep Ensembles in terms of predictive performance and training data-efficiency and prevents overconfidence and posterior collapse.

## 2  RELATED WORK

As the exact posterior over all weights of a neural network is intractable, approximate methods are essential for scalable BNNs. These methods generally fall into two categories: sampling-based approaches and those that approximate the posterior with parameterized distributions.

**Markov Chain Monte Carlo** (MCMC) methods attempt to draw representative samples from the posterior distribution over weights. Although methods such as Hamiltonian Monte Carlo are asymptotically exact, they become computationally prohibitive for large NNs due to their high-dimensional parameter spaces and complex energy landscapes (Coker et al., 2022). An adaptation of Gibbs sampling has been scaled to MNIST, but on a very small network with only 8,180 parameters (Papamarkou, 2023). Approximate sampling methods can be faster but still require a large number of samples, which complicates both training and inference. Although approaches like knowledge distillation (Korattikara et al., 2015) attempt to speed up inference, MCMC methods remain too inefficient for large-scale deep learning applications (Khan & Rue, 2024).

**Variational Inference** (VI) aims to approximate the intractable posterior distribution $p(\boldsymbol{w}_1, \ldots, \boldsymbol{w}_K \,|\, \mathcal{D})$ by a variational posterior $q(\boldsymbol{w}_1, \ldots, \boldsymbol{w}_K)$. The parameters of $q$ are optimized using gradients with respect to an objective function, which is typically a generalized form of the reverse KL divergence $D_{\text{KL}} \left[ q(\boldsymbol{w}_1, \ldots, \boldsymbol{w}_K) \,\|\, p(\boldsymbol{w}_1, \ldots, \boldsymbol{w}_K \,|\, \mathcal{D}) \right]$. Early methods like (Graves, 2011) and Bayes By Backprop (Blundell et al., 2015) laid the foundation for applying VI to NNs, but suffer from slow convergence and severe underfitting, especially for large models or small dataset sizes (Osawa et al., 2019). More recently, VOGN (Osawa et al., 2019) achieved Adam-like results on ImageNet LSVRC by applying a Gauss-Newton approximation to the Hessian matrix. IVON (Shen et al., 2024) improved upon VOGN by using cheaper Hessian approximations and training techniques such as gradient clipping, achieving Adam-like performance on large-scale models such as GPT-2 while maintaining similar runtime costs. Despite recent advances, VI continues to face challenges such as overconfidence, posterior collapse, and complex hyperparameter tuning, motivating the exploration of alternative approaches (Zhang et al., 2018).

**Message Passing (MP) for Neural Networks**: MP is a general framework that unifies several algorithms (Kschischang et al., 2001; Minka, 2001), but its direct application to NNs has been limited. Expectation backpropagation (EBP) (Soudry et al., 2014) approximates the posterior of 3-layer MLPs by combining expectation propagation, an approximate MP algorithm, with gradient backpropagation. Similarly, probabilistic backpropagation (PBP) (Hernández-Lobato & Adams, 2015) combines belief propagation with gradient backpropagation and was found to produce better approximations than EBP (Ghosh et al., 2016). However, PBP treats the data as new examples in each consecutive epoch (double-counting), which makes it prone to overconfidence. Furthermore, both EBP and PBP only deployed on small datasets and rely on gradients instead of pure MP. In contrast, Lucibello et al. (2022) applied MP to larger architectures by modeling the posterior over NN weights as a factor graph, but faced posterior collapse to a point measure due to also double-counting data. Their experiments were mostly restricted to three-layer MLPs without biases and with binary weights. Our approach builds on this by introducing an MP framework for BNNs that avoids double-counting, scales to CNNs, and effectively supports continuous weights.

## 3 Theoretical Model

Our goal is to model the predictive posterior of a BNN as a factor graph and find a Gaussian approximation of the predictive posterior via belief propagation. Essentially, factor graphs are probabilistic modelling tools for approximating the marginals of joint distributions, provided that they factorize sufficiently. For a more comprehensive introduction on factor graphs and the sum-product algorithm, refer to Kschischang et al. (2001) BNNs, on the other hand, treat the parameters $\theta$ of a model $f_\theta : \mathbb{R}^d \longrightarrow \mathbb{R}^o$ as random variables with prior beliefs $p(\theta)$. Given a training dataset $\mathcal{D} = \{\boldsymbol{x}_i, \boldsymbol{y}_i\}_{i=1}^n$ of i.i.d. samples, a likelihood relationship $p(\boldsymbol{y} \,|\, \boldsymbol{x}, \theta) = p(\boldsymbol{y} \,|\, f_\theta(\boldsymbol{x}))$, and a new input sample $\boldsymbol{x}$, the goal is to approximate the predictive posterior distribution $p(\boldsymbol{y} \,|\, \boldsymbol{x}, \mathcal{D})$, which can be written as:

$$p(\boldsymbol{y} \,|\, \boldsymbol{x}, \mathcal{D}) = \int p(\boldsymbol{y} \,|\, \boldsymbol{x}, \theta) \, p(\theta \,|\, \mathcal{D}) \, d\theta. \tag{1}$$

This means that the density of the predictive posterior is the expected likelihood under the posterior distribution $p(\theta \,|\, \mathcal{D})$, which is proportional[1] to the product of the prior and dataset likelihood:

$$p(\theta \,|\, \mathcal{D}) \propto p(\theta) \prod_{i=1}^{n} p(\boldsymbol{y}_i \,|\, f_\theta(\boldsymbol{x})). \tag{2}$$

The integrand in Equation (1) exhibits a factorized structure that is well-suited to factor graph modeling. However, directly modelling the relationship $\boldsymbol{o} = f_\theta(\boldsymbol{x})$ with a single Dirac delta factor $\delta(\boldsymbol{o} - f_\theta(\boldsymbol{x}))$ does not yield feasible message equations. Therefore, we model the NN at scalar level by introducing intermediate latent variables connected by elementary Dirac delta factors. Figure 1 illustrates this construction for a simple MLP with independent weight matrices a priori. While the abstract factor graph in the figure uses vector variables for simplicity, we actually derive message equations where each vector component is treated as a separate scalar variable and all Dirac deltas depend only on scalar variables. For instance, if $d = 2$, the conceptual factor $\delta(\mathbf{o} - \mathbf{W}_2 \mathbf{a})$ is replaced by four scalar factors: $\delta(\mathrm{p}_{jk} - \mathrm{w}_{jk}\mathrm{a}_k)$ for $j, k = 1, 2$, with intermediate variables $\mathrm{p}_{jk}$, and two factors $\delta(\mathrm{o}_j - (\mathrm{p}_{j1} + \mathrm{p}_{j2}))$. By multiplying all factors in this expanded factor graph and integrating over intermediate results, we obtain a function in $\boldsymbol{x}, \boldsymbol{y}, \theta$ that is proportional to the integrand in Equation (1). Hence, the marginal of the unobserved target $\mathbf{y}$ is proportional to $p(\boldsymbol{y} \,|\, \boldsymbol{x}, \mathcal{D})$. When $\mathbf{y}$ connects to only one factor, its marginal matches its incoming message.

## 4 Approximations

Calculating a precise representation of the message to the target of an unseen input is intractable for large networks and datasets. The three primary reasons are, that a) nonlinearities and multiplication produce highly complex exact messages which are difficult to represent and propagate, b) the enormous size of the factor graph for large datasets, and c) the presence of various cycles in the graph. These challenges shape the message approximations as well as the design of our training and prediction procedures, which we address in the following sections.

---

[1]with a proportionality constant of $1/p(\mathcal{D})$

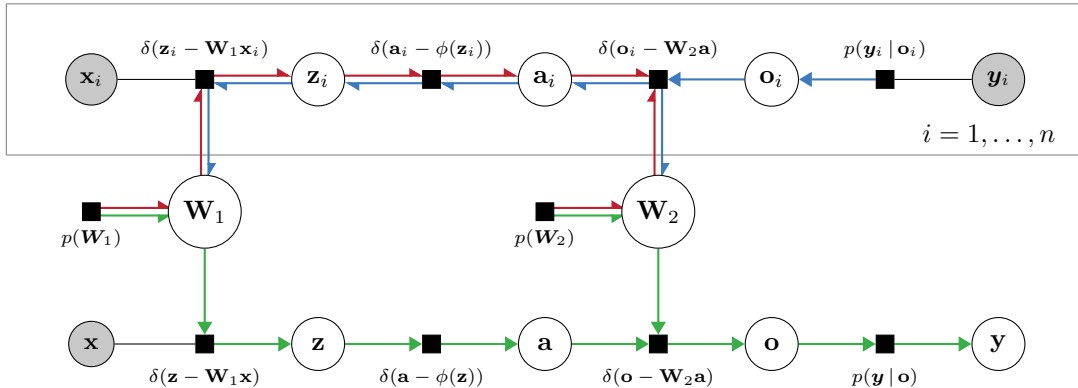

Figure 1: Conceptual vector-valued factor graph for a simple MLP. Each training example has its own "branch" (a copy of the network), while predictions for an unlabeled input $x$ are computed on a separate prediction branch. All branches are connected by the shared model parameters. Grayed-out variables are conditioned on (observed). Colored arrows indicate the three iteration orders: a forward / backward pass on training examples, and a forward pass for prediction. For more details, we refer to the documentation contained in the supplement of the paper.

## 4.1 APPROXIMATING MESSAGES VIA GAUSSIAN DENSITIES

To work around the highly complex exact messages, we approximate them with a parameterized class of functions. We desire this class to be closed under pointwise multiplication, as variable-to-factor messages are the product of incoming messages from other factors. We choose positive scalar multiples of one-dimensional Gaussian densities as our approximating family. Their closedness follows immediately from the exponential function's characteristic identity $\exp(x)\exp(y) = \exp(x+y)$ and the observation that for any $s_1, s_2 > 0$ and $\mu_1, \mu_2 \in \mathbb{R}$, the function $s_1(x-\mu_1)^2 + s_2(x-\mu_2)^2$ in $x$ can be represented as $s(x-\mu)^2 + c$ for some $s > 0$ and $\mu, c \in \mathbb{R}$. The precise relation between two scaled Gaussian densities and its product can be neatly expressed with the help of the so-called natural (re-)parameterization. Given a Gaussian $\mathcal{N}(\mu, \sigma^2)$, we call $\rho = 1/\sigma^2$ the precision and $\tau = \mu/\sigma^2$ the precision-mean. Collectively, $(\tau, \rho)$ are the Gaussian's natural parameters, $\mathbb{G}(x; \tau, \rho) := \mathcal{N}(x; \mu, \sigma^2)$, $x \in \mathbb{R}$. For $\mu_1, \mu_2 \in \mathbb{R}$ and $\sigma_1, \sigma_2 > 0$ with corresponding natural parameters $\rho_i = 1/\sigma_i^2$ and $\tau_i = \mu_i \rho_i$, $i = 1, 2$, multiplying Gaussian densities simplifies to:

$$\mathcal{N}(\mu_1; \mu_2, \sigma_1^2 + \sigma_2^2) \cdot \mathbb{G}(x; \tau_1 + \tau_2, \rho_1 + \rho_2)$$

for all $x \in \mathbb{R}$. Thus, multiplying Gaussian densities simplifies to the pointwise addition of their natural parameters, aside from a multiplicative constant. Since we are only interested in the marginals, which are re-normalized, this constant does not affect the final result.

Next, we present our message approximations for three factor types, each representing a deterministic relationship between variables: (i) the sum of variables weighted by constants, (ii) the application of a nonlinearity, and (iii) the multiplication of two variables. As we model the factor graph on a scalar level, these three factors suffice to model complex modern network architectures such as ConvNeXt Liu et al. (2022)[2]. In Appendix F, we provide a comprehensive table of message equations, including additional factors for modeling training labels.

**Weighted Sum**: The density transformation property of the Dirac delta allows us to compute the exact message without approximation. For the relationship $s = c^\intercal v$ modeled by the factor $f := \delta(s - c^\intercal v)$,

$$m_{f \to s}(s) = \int \delta(s - c^\intercal v) \prod_{i=1}^{k} m_{v_i \to f}(v_i) \, dv_1 \dots v_k$$

is simply the density of $c^\intercal v$, where $\mathbf{v} \sim \prod_{i=1}^{k} m_{v_i \to f}(v_i)$. If $m_{v_i \to f}(v_i) = \mathcal{N}(v_i; \mu_i, \sigma_i^2)$ are Gaussian, then $\mathbf{v} \sim \mathcal{N}(\boldsymbol{\mu}, \text{diag}(\boldsymbol{\sigma}^2))$ and $m_{f \to s}(s)$ becomes a scaled multivariate Gaussian:

$$m_{f \to s}(s) = \mathcal{N}(s; c^\intercal \boldsymbol{\mu}, (c^2)^\intercal \boldsymbol{\sigma}^2).$$

---

[2]with the exception of layer normalization, which can be substituted by orthogonal initialization schemes Xiao et al. (2018) or specific hyperparameters of a corresponding normalized network Nguyen et al. (2023)

The backward messages $m_{f \to v_i}$ can be derived similarly without approximation.

**Nonlinearity**: We model the application of a nonlinearity $\phi : \mathbb{R} \to \mathbb{R}$ as a factor $f := \delta(a - \phi(z))$. However, the forward and backward messages are problematic and require approximation–even for well-behaved, injective $\phi$ such as $\text{LeakyReLU}_\alpha$:

$$m_{a \to f}(a) = \text{pdf}_{\phi(Z)}(a) \text{ for } Z \sim \mathcal{N}$$

$$m_{f \to z}(z) = \int \delta(a - \phi(z)) \cdot m_{a \to f}(a) \, da = m_{a \to f}(\phi(z)) = \mathcal{N}(\phi(z); \mu_a, \sigma_a^2).$$

For values of $\alpha \neq 1$, the forward message is non-Gaussian and the backward message does not even integrate to 1. For ReLU ($\alpha = 0$), it is clearly not even integrable. Instead, we use *moment matching* to fit a Gaussian approximation. Given any factor $f$ and variable v, we can approximate the message $m_{f \to v}$ directly with a Gaussian if the moments $m_k := \int v^k m_{f \to v}(v) \, dv$ exist for $k = 0, 1, 2$ and can be computed efficiently via $m_{f \to v}(v) = \mathcal{N}(v; m_1/m_0, m_2/m_0 - (m_1/m_0)^2)$. However, direct moment matching of the message is impossible for non-integrable messages or when the $m_k$ are expensive to find. Instead, we can apply moment matching to the updated *marginal* of v. Let $m_0$, $m_1$, $m_2$ be the moments of the "true" marginal

$$m(v) = \int f(v, v_1, ..., v_k) \, dv_1...dv_k \cdot \prod_i m_{g_i \to v}(v),$$

which is the product of the true message from $f$ and the approximated messages from other factors $g_i$. Then we can approximate $m$ with a Gaussian and obtain a message approximation

$$m_{f \to v}(v) := \mathcal{N}(v; \mu_v, \sigma_v^2)/m_{v \to f}(v),$$

which approximates $m_{f \to v}$ so that it changes v's marginal in the same way as the actual message.[3] Since $m_{v \to f}(v)$ is a Gaussian density, we can compute $m_{f \to v}(v)$ efficiently by applying Gaussian division in natural parameters, similar to Section 4.1. For $\text{LeakyReLU}_\alpha$, we found efficient direct and marginal approximations that are each applicable to both the forward and backward message when $\alpha \neq 0$. The marginal approximation remains applicable even for the ReLU case of $\alpha = 0$. We provide detailed derivations in Appendix B.2.

**Product** For the relationship c = ab, we employ variational MP as in Stern et al. (2009), in order to break the vast number of symmetries in the true posterior of a BNN. By combining the variational message equations for scalar products with the weighted sum, we can also construct efficient higher-order multiplication factors such as inner vector products, see Appendix F for detailed equations.

## 4.2 TRAINING PROCEDURE & PREDICTION

In pure belief propagation, the product of incoming messages for any variable equals its marginal under the true posterior. With our aforementioned approximations, we can reasonably expect to converge on a diagonal Gaussian $\check{q}$ that approximates one of the various permutation modes of the true posterior by aligning the first two moments of the marginal. This concept can be elegantly interpreted through the lens of relative entropy. As shown in A.2, among diagonal Gaussians $q(\theta) = q_1(\theta_1) \cdots q_k(\theta_k)$, the relative entropy from the true posterior to $q$ is minimized for $\check{q}$:

$$\check{q} = \text{argmin}_q D_{\text{KL}}\left[\, p(\theta \,|\, \mathcal{D}) \,|\, q(\theta)\,\right]. \tag{3}$$

Another challenge in finding $\check{q}$ arises from cyclic dependencies. In acyclic factor graphs, each message depends only on previous messages from its subtree, allowing for exact propagation. However, our factor graph contains several cycles due to two primary reasons: (i) multiple training branches interacting with shared parameters across linear layers, and (ii) the scalar-level modeling of matrix-vector multiplication in architectures with more than one hidden layer. These loops create circular dependencies among messages. To address these challenges, we adopt loopy belief propagation, where belief propagation is performed iteratively until messages converge. While exact propagation works in acyclic graphs, convergence is then only guaranteed under certain conditions (e.g., Simon's condition (Ihler et al., 2005)) that are difficult to verify. Instead, we pass messages in an iteration order that largely avoids loops by alternating forward and backward passes similarly to deterministic NNs. Our message schedule is visualized in Figure 1.

---

[3]This is the central idea behind expectation propagation as defined in Minka (2001).

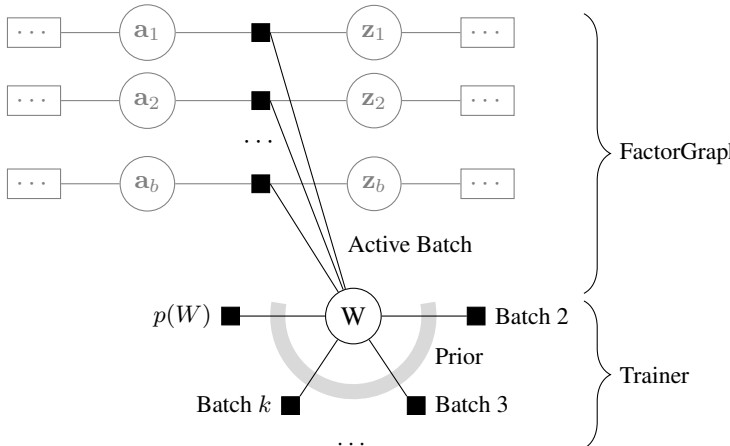

Figure 2: A full FactorGraph models all messages for one batch of training examples. To iterate, we only need one joint message summarizing the prior and all other examples. When switching to a new batch, we aggregate messages from the previous batch and store them in the Trainer.

**Batching:** As the forward and backward messages depend on each other, we must store them to compute message updates during message passing. Updating our messages in a sweeping "pass" over a branch and running backward passes immediately after the forward pass on the same branch, allows us to store many messages only temporarily, reducing memory requirements. This schedule also ensures efficient propagation of updated messages despite the presence of loops. However, some messages must still be retained permanently[4], leading to significant memory demand when storing them for all $n$ training examples. To address this, we adopt a batching strategy: Instead of maintaining $n$ training branches simultaneously, we update the factor graph using a batch (subset) of $b$ examples at a time. The factor graph then models $b$ messages to the weights $W$, while the messages to $W$ from the remaining (inactive) examples are aggregated into batch-wise products and stored in a trainer object. Figure 2 illustrates this setup. When switching batches, we divide the marginals by the batch's old aggregate message and multiply the updated messages into the marginal, ensuring that data is not double-counted. Within each batch, we iterate through the examples and perform a forward and backward pass on each in sequence. After all examples have been processed once, we call it an "iteration". Depending on the training stage, we either repeat this process within the same batch or move to the next batch. As training progresses, we gradually increase the number of iterations per batch to allow for finer updates as the overall posterior comes closer to convergence.

**Prediction:** Ultimately, our goal is to compute the marginal of the unobserved target $\mathbf{y}$ for some unseen input $\mathbf{x}$. Since the prediction branch in Figure 1 introduces additional loops, obtaining an accurate approximation would require iterating over the entire factor graph, including the training branches. In NN terms, this translates to retraining the whole network for every test input. Instead, we pass messages only on the training branches in the batch-wise setup described above. At test time, messages from the training branches are propagated to the prediction branch, but not vice versa. Specifically, messages from the weights to the prediction branch are computed as the product of the prior and the incoming messages from the training branches. This can be interpreted as approximating the posterior over weights, $p(\theta \,|\, \mathcal{D})$, with a diagonal Gaussian $\check{q}(\theta)$ used as prior during inference.

### 4.3 IMPLEMENTATION

Scaling the approach to deep networks, the following challenges need to be addressed.

**Factor Graph Implementation** While batching effectively reduces memory requirements for large datasets, a direct implementation of a factor graph still scales poorly for deep networks. Explicitly modeling each scalar variable and factor as an instance is computationally expensive. To address this, we propose the following design optimizations: First, rather than modeling individual elements of the factor graph, we represent entire layers of the network. MP between layers is orchestrated by

---

[4]For example, the backward message of the linear layer is needed to compute the marginal of the inputs, which the forward message depends on.

an outer training loop. Second, each layer instance operates across all training branches within the active batch, removing the need to duplicate layers for each example. Third, factors are stateless functions, not objects. Each layer is responsible for computing its forward and backward messages by calling the required functions. In this design, layer instances maintain their own state, but MP and batching are managed in the outer loop. The stateless message equations are optimized for both performance and numerical stability. As a result, the number of layer instances scales linearly with network depth but remains constant regardless of layer size or batch size. This approach significantly reduces computational and memory overhead—our implementation is approximately 300x faster than a direct factor graph model in our tests. Additionally, we optimized our implementation for GPU execution by leveraging Julia's `CUDA.jl` and `Tullio.jl` libraries. Since much of the runtime is spent on linear algebra operations (within linear or convolutional layers), we built a reusable, GPU-compatible library for Gaussian multiplication. This design makes the implementation both scalable and extendable.

**Numerical Stability**    Maintaining numerical stability in the MP process is critical, particularly as model size increases. Backward messages often exhibit near-infinite variances when individual weights have minimal impact on the likelihood. Therefore, we compute them directly in natural parameters, which also simplifies the equations. Special care is needed for LeakyReLU, as its messages can easily diverge. To mitigate this, we introduced guardrails: when normalization constants become too small, precision turns negative, or variance in forward messages increases, we revert to either $\mathbb{G}(0,0)$ or use moment matching on messages instead of marginals (see Appendix F for details). Another trick is to periodically recompute the weight marginals from scratch to maintain accuracy. By leveraging the properties of Gaussians, we save memory by recomputing variable-to-factor messages as needed[5]. However, incremental updates to marginals can accumulates errors, so we perform a full recomputation once per batch iteration. Lastly, we apply light message damping through an exponential moving average to stabilize the training, but, importantly, only on the aggregated batch messages, not on the individual messages of the active batch.

**Weight Priors**    A zero-centered diagonal Gaussian prior with variance $\sigma_p^2$ is a natural choice for the prior over weights. However, as in traditional deep learning, setting all means to zero prevents messages from breaking symmetry. To resolve this, we sample prior means according to spectral parametrization (Yang et al., 2024), which facilitates feature learning independent of the network width. Another challenge in prior choice is managing exploding variances. In a naive setup with $\sigma_p^2 = 1$, forward message variances grow exponentially with the network depth. To find a principled choice of $\sigma_p^2$, our initialization scheme is based on experimental data, see Appendix D.

## 5   NUMERICAL EVALUATION

**Experiment 1: Application on MNIST dataset.**    In our experiments on the MNIST dataset, we compare regression and classification-based versions of our message passing (MP) and SGD. Table 1 compares the test accuracy of MP and SGD for 3-layer MLPs and the LeNet-5 architecture (Lecun et al., 1998) over a range of training set sizes. We found that R-MP is generally more effective than AM-MP and that both consistently yield better accuracy than SGD, in particular for limited training data. For instance, our regression-based MP (R-MP) achieves 85.69% accuracy on the MLP with only 640 training samples, significantly outperforming softmax-based SGD's (SM-SGD) 58.85%. We also trained a 3-layer MLP of width 2,000 with 5.6 million parameters, which reached a test accuracy of 98.04%, whereas at width 256 the accuracy was 98.33%. Among related work on message passing, results for MNIST-sized datasets were only published by Lucibello et al. (2022). Their method reached only 97.4% test accuracy and they published no metrics for evaluating their predictive uncertainty. For VI, Bayes By Backprop reported an accuracy of 98.18% for their Gaussian model and 98.64% for their mixture model, which are similar to the accuracy achieved by MP.

A key strength of our approach lies in the performance of its predictive uncertainty. Figure 3a shows that for a training dataset of size 640, counterintuitively, SGD is underconfident (ECE of 0.3695) whereas R-MP and AM-MP are both decently calibrated with an ECE of 0.0216 and 0.0251 respectively. All methods achieve good calibration when trained on the whole training data, with calibration errors of 0.0019 for SGD, 0.0014 for AM-MP, and 0.001 for R-MP. However, since most

---

[5]Each layer stores factor-to-weight-variable messages and the marginal, which is an aggregate that is continuously updated as individual messages change.

|  | Num Data | 80 | 160 | 320 | 640 | 1 280 | 2 560 | 5 120 | 10 240 | 60 000 |
|---|---|---|---|---|---|---|---|---|---|---|
| **MLP** | **R-MP** | 30.01 | **61.79** | 77.61 | **85.69** | **88.95** | **91.72** | **94.85** | **96.25** | **98.33** |
| | **AM-MP** | **31.88** | 61.08 | **80.79** | 85.50 | 87.92 | 91.56 | 94.08 | 95.72 | 98.21 |
| | **R-SGD** | 10.11 | 11.45 | 14.89 | 29.66 | 49.78 | 67.01 | 76.41 | 83.00 | 92.22 |
| | **SM-SGD** | 21.47 | 30.38 | 46.18 | 58.83 | 76.67 | 85.55 | 89.10 | 91.17 | 96.36 |
| **LeNet-5** | **R-MP** | **27.75** | **25.58** | **38.02** | **94.72** | 95.36 | 96.32 | 97.40 | **98.12** | **99.02** |
| | **AM-MP** | 17.32 | 10.42 | 10.28 | 93.48 | **96.19** | **96.44** | **97.70** | 98.05 | 98.95 |
| | **R-SGD** | 14.06 | 14.51 | 14.07 | 13.99 | 16.02 | 31.16 | 49.43 | 69.84 | 94.12 |
| | **SM-SGD** | 18.57 | 19.54 | 21.03 | 22.15 | 39.36 | 82.30 | 90.92 | 95.04 | 98.55 |

Table 1: Comparison of accuracies on MNIST (% correct). Our method (MP) consistently achieves better accuracy than SGD (Torch). Abbreviations: Regression (R), Argmax (AM), and Softmax (SM).

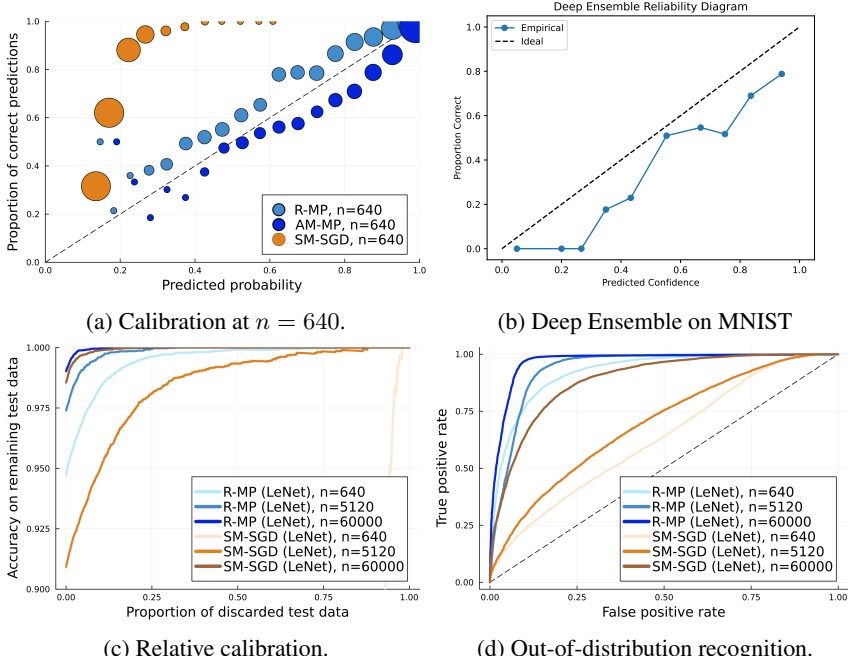

(a) Calibration at $n = 640$.

(b) Deep Ensemble on MNIST

(c) Relative calibration.

(d) Out-of-distribution recognition.

Figure 3: Uncertainty metrics for models trained on MNIST.

examples have high confidence levels, calibration becomes less informative at larger dataset sizes. Thus, we employ relative calibration curves to assess uncertainty further[6]. Figure 3b illustrates the predicted confidence for Deep Ensembles. We observe that Deep Ensembles tend to be overconfident. Figure 3c compares the relative calibration of R-MP and SM-SGD for LeNet-5. Overall, the R-MP predictions show excellent relative calibration with an area under the curve (AUC) of 0.9949, 0.9986, 0.9998 for 640, 5 120, and 60 000 datapoints, whereas SM-SGD only achieved 0.4451, 0.9845, and 0.9995 respectively. Finally, we evaluated out-of-distribution (OOD) recognition by training a model on MNIST and then predicting on mixed examples from FashionMNIST and MNIST. Figure 3d shows the receiver operating characteristic (ROC) curve for detecting OOD samples by the entropy of their predicted class distribution. R-MP achieved an AUC of 0.9675 when trained on full MNIST and 0.9242 for $n = 640$, whereas SM-SGD only achieved 0.8872 even with the full training data.

**Experiment 2: Application on CIFAR-10.**    To evaluate the applicability of our method on the CIFAR-10 dataset, we trained a 6 layer deep convolutional network with roughly 890k parameters on the full training dataset. As baseline methods we used the SOTA optimizers AdamW (Loshchilov & Hutter, 2017) and IVON (Shen et al., 2024) each with a cosine annealing learning rate schedule

---

[6]We order the test examples by their predicted max-class probability. For each uncertainty cutoff, we then plot the accuracy on the remaining (more certain) test set. The area under this curve is also reported under the name AUROC by Osawa et al. (2019).

|  | Acc. ↑ | Top-5 Acc. ↑ | NLL ↓ | ECE ↓ | Brier ↓ | OOD-AUROC ↑ |
|---|---|---|---|---|---|---|
| AdamW | **0.783** | **0.984** | 1.736 | 0.046 | 0.380 | 0.792 |
| IVON@mean | 0.772 | 0.983 | 1.494 | 0.041 | 0.387 | **0.819** |
| IVON | 0.772 | 0.983 | 1.316 | 0.035 | 0.370 | 0.808 |
| Deep Ensemble | 0.819 | 0.987 | **0.689** | 0.032 | **0.257** | 0.767 |
| MP (Ours) | 0.773 | 0.977 | 0.997 | **0.029** | 0.361 | 0.810 |

Table 2: Comparison of various validation statistics for a convolutional network of roughly 890k parameters trained on CIFAR-10. Out-of-distribution (OOD) detection was tested with SVHN. For IVON we used 100 samples for prediction at test time. IVON@mean are the results obtained from evaluating the model at the means of the learned distributions of the individual parameters. For deep ensemble, 30 independent models of AdamW were executed. Abbreviations: Acc. = Accuracy, NLL = Normalized Log-Likelihood, ECE = Expected Calibration Error, Brier = Brier score.

(Loshchilov & Hutter, 2016). Across all methods, including ours, we trained for 25 epochs. In Appendix C, we give extensive details on the network architecture and the experimental setup in general. Table 2 compares the performance of our method (MP) against AdamW and IVON across a variety of standard metrics. In general, we see that MP can compete with these two strong baselines. In the expected calibration error, our method even has a notable edge. The fact that the metrics are overall worse than what is reported by Shen et al. (2024) is probably due to a difference in architecture; Shen et al. only conduct experiments on ResNets equipped with filter response normalization (Singh & Krishnan, 2019). Neither residual connections nor normalization layers are yet implemented in our factor graph library. Nevertheless, the potential of the approach becomes already visible.

**Experiment 3: Further Evaluations on Tabular Benchmark Data.** We use the UCI machine learning repository, cf. Dua & Graff (2017), for various regression tasks. The results, see Appendix E.1, show that our method is general applicable and effectively avoids overfitting.

## 6 SUMMARY, LIMITATIONS & FUTURE WORK

**Summary** We presented a novel framework that advances message-passing (MP) for BNNs by modeling the predictive posterior as a factor graph. To the best of our knowledge, this is the first MP method to handle CNNs while avoiding double-counting training data, a limitation in previous MP approaches like Soudry et al. (2014); Hernández-Lobato & Adams (2015); Lucibello et al. (2022). In our experiments on different datasets, our method proved to be competitive with the SOTA baselines AdamW and IVON, while – as intended – showing clearly better uncertainty quantification.

**Limitations** Despite recent advances, VI methods like IVON remain ahead in scale and performance on larger datasets. Our approach's runtime and memory requirements scale linearly with model parameters and dataset size. While our inference at test time can keep up with IVON's sampling approach in terms of speed and memory requirements, training is up to two orders of magnitude slower and more GPU-memory intensive compared to training deterministic networks using PyTorch with optimizers like AdamW. The memory overhead stems from two key factors: First, each training example stores messages proportional to the model's parameter count, unlike AdamW's batch-level intermediate representations. Second, each parameter requires two 8-byte floating-point numbers, contrasting with more efficient 4-byte or smaller formats. Runtime inflation results from several performance bottlenecks: Our training schedule lacks parallel forward passes, our Tullio-based CUDA kernel generation misses memory-layout and GPU optimizations present in mature libraries like Torch, message equations involve complex computations beyond standard matrix multiplications, and we use Julia's default FP64 precision, which GPUs process less efficiently.

**Future Work** Regarding training efficiency, an altered message-update schedule with actual batched computations would significantly reduce training time. Implementing our library in CUDA C++ with efficiency in mind could also drastically cut down computational overhead. On the architectural front, we deem it likely that our approach can be extended to most modern deep learning architectures. Residual connections are straightforward to implement as they boil down to simple sum factors. For normalization layers at the scalar level, only a division factor is missing, which can be approximated by a "rotated" product factor. This would suffice to model ResNet-like architectures and more modern convolutional networks like ConvNeXt.

**Reproducibility** All code is available at https://github.com/neurips-submission-19866/submission.

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

## A  PROOF OF GLOBAL MINIMIZATION OBJECTIVE

### A.1  MOMENT-MATCHED GAUSSIANS MINIMIZE CROSS-ENTROPY

Consider a scalar density $p$ and a Gaussian $q(\theta) = \mathcal{N}(\theta, \mu, \sigma)$. Then

$$\min H(p, q) = \min \left( \int p(\theta) \log \left( \frac{p(\theta)}{q(\theta)} \right) d\theta \right) = \min \left( \frac{1}{2\sigma^2} \int p(\theta)(\theta - \mu)^2 \, d\theta + \frac{\log(2\pi\sigma^2)}{2} \right).$$

It is well known that expectations minimize the expected mean squared error. In other words, the integral is minimized by setting $\mu$ to the expectation of $p$ and is then equal to the variance of $p$. The necessary condition of a local minimum then yields that $\sigma^2$ must be the variance of $p$.

### A.2  PROOF OF EQUATION (3) GLOBAL MINIMIZATION OBJECTIVE

Let $p$ be an arbitrary probability density on $\mathbb{R}^k$ with marginals $p_i(\theta_i) := \int p(\theta) \, d(\theta \setminus \theta_i)$ and denote by $\mathcal{Q}$ the set of diagonal Gaussians. Then for every $q(\boldsymbol{\theta}) = \prod_{i=1}^k q_i(\theta_i) \in \mathcal{Q}$ we can write the relative entropy from $p$ to $q$ as

$$D_{\mathrm{KL}}[\, p \,||\, q\,] = \int p(\boldsymbol{\theta}) \log \left( \frac{p(\boldsymbol{\theta})}{q(\boldsymbol{\theta})} \right) d\boldsymbol{\theta} = -\sum_{i=1}^k \int p(\boldsymbol{\theta}) \log(q(\theta_i)) d\boldsymbol{\theta} - H(p)$$

$$= -\sum_{i=1}^k \int_{\theta_i} \log(q_i(\theta_i)) \int_{\boldsymbol{\theta} \setminus \theta_i} p(\boldsymbol{\theta}) d(\boldsymbol{\theta} \setminus \theta_i) - H(p) = \sum_{i=1}^k H(p_i, q_i) - H(p).$$

This shows that $D_{\mathrm{KL}}[\, p \,||\, q\,]$ is minimized by independently minimizing the summands $H(p_i, q_i)$. In combination with A.1 this completes the proof.

## B  DERIVATIONS OF MESSAGE EQUATIONS

### B.1  RELU

A common activation function is the Rectified Linear Unit $\mathrm{ReLU} : \mathbb{R} \to \mathbb{R}, z \mapsto \max(0, z)$.

**Forward Message:**  Since ReLU is not injective, we cannot apply the density transformation property of the Dirac delta to the forward message

$$m_{f \to \mathrm{a}}(\mathrm{a}) = \int_{\mathrm{z} \in \mathbb{R}} \delta(\mathrm{a} - \mathrm{ReLU}(\mathrm{z})) m_{\mathrm{z} \to f}(\mathrm{z}) \, d\mathrm{z}.$$

In fact, the random variable $\mathrm{ReLU}(Z)$ with $Z \sim m_{z \to f}$ does not even have a density. A positive amount of weight, namely $\Pr[Z \leq 0]$, is mapped to 0. Therefore

$$m_{f \to \mathrm{a}}(0) = \lim_{t \to 0} \int_{\mathrm{z} \in \mathbb{R}} \mathcal{N}(\mathrm{ReLU}(\mathrm{z}); 0, t^2) m_{\mathrm{z} \to f}(\mathrm{z}) \, d\mathrm{z} \geq \lim_{t \to 0} \mathcal{N}(0; 0, t^2) \min_{\mathrm{z} \in [-1, 0]} m_{\mathrm{z} \to f}(\mathrm{z}) = \infty.$$

Apart from 0, the forward message is well defined everywhere, and technically null sets do not matter under the integral. However, moment-matching $m_{z \to f}$ while truncating at 0 does not seem reasonable as it completely ignores the weight of $m_{z \to f}$ on $\mathbb{R}_{\leq 0}$. Therefore, we refrain from moment-matching the forward message of ReLU.

As an alternative, we consider a marginal approximation. That means, we derive formulas for

$$m_k := \int_{\mathrm{a} \in \mathbb{R}} \mathrm{a}^k m_{\mathrm{a} \to f}(\mathrm{a}) m_{f \to \mathrm{a}}(\mathrm{a}) \, d\mathrm{a}, \quad k \in \{0, 1, 2\} \tag{4}$$

and then set

$$m_{f \to \mathrm{a}}(\mathrm{a}) := \mathcal{N}\left(\mathrm{a}; m_1/m_0, m_2/m_0 - (m_1/m_0)^2\right) / m_{\mathrm{a} \to f}(\mathrm{a}).$$

By changing the integration order, we obtain

$$m_k = \int_{a \in \mathbb{R}} a^k m_{a \to f}(a) \int_{z \in \mathbb{R}} \delta(a - \text{ReLU}(z)) m_{z \to f}(z) \, dz \, da$$

$$= \int_{z \in \mathbb{R}} m_{z \to f}(z) \int_{a \in \mathbb{R}} \delta(a - \text{ReLU}(z)) a^k m_{a \to f}(a) \, da \, dz$$

$$= \int_{z \in \mathbb{R}} m_{z \to f}(z) \text{ReLU}^k(z) m_{a \to f}(\text{ReLU}(z)) \, dz$$

Note that we end up with a well-defined and finite integral. Similar integrals arise in later derivations. For this reason we encapsulate part of the analysis in basic building blocks.

**Building Block 1** *We can efficiently approximate integrals of the form*

$$\int_0^\infty z^k \mathcal{N}(z; \mu_1, \sigma_1^2) \mathcal{N}(z; \mu_2, \sigma_2^2) \, dz$$

*where $\mu_1, \mu_2 \in \mathbb{R}, \sigma_1, \sigma_2 > 0$ and $k = 0, 1, 2$.*

**ESONG PROOF 1** *By Section 4.1 the integral is equal to*

$$S^+ = \mathcal{N}(\mu_1; \mu_2, \sigma_1^2 + \sigma_2^2) \int_0^\infty z^k \mathcal{N}\left(z; \mu, \sigma^2\right) \, dz$$

$$= \mathcal{N}(\mu_1; \mu_2, \sigma_1^2 + \sigma_2^2) \begin{cases} \mathbb{E}[ReLU^k(\mathcal{N}(\mu, \sigma^2))] & \text{for } k = 1, 2 \\ \Pr[-Z \le 0] = \phi(\mu/\sigma) & \text{for } k = 0 \end{cases}$$

*where*

$$\mu = \frac{\tau}{\rho}, \quad \sigma^2 = \frac{1}{\rho}, \quad \tau = \frac{\mu_1}{\sigma_1^2} + \frac{\mu_2}{\sigma_2^2} \quad \text{and} \quad \rho = \frac{1}{\sigma_1^2} + \frac{1}{\sigma_2^2}.$$

This motivates the derivation of efficient formulas for the moments of an image of a Gaussian variable under ReLU.

**Building Block 2** *Let $Z \sim \mathcal{N}(\mu, \sigma^2)$. The first two moments of $ReLU(Z)$ are then given by*

$$\mathbb{E}[ReLU(Z)] = \sigma\varphi(x) + \mu\phi(x) \tag{5}$$

$$\mathbb{E}[ReLU^2(Z)] = \sigma\mu\varphi(x) + (\sigma^2 + \mu^2)\phi(x), \tag{6}$$

*where $x = \mu/\sigma$ and $\varphi, \phi$ denote the pdf and cdf of the standard normal distribution, respectively.*

**ESONG PROOF 2** *The basic idea is to apply $\int z e^{-z^2/2} \, dz = -e^{-z^2/2}$. Together with a productive zero, one obtains*

$$\sqrt{2\pi}\sigma\mathbb{E}[ReLU(Z)] = \int_0^\infty z e^{-\frac{(z-\mu)^2}{2\sigma^2}} \, dz = \sigma^2 \int_0^\infty \frac{(z-\mu)}{\sigma^2} e^{-\frac{(z-\mu)^2}{2\sigma^2}} \, dz + \mu \int_0^\infty e^{-\frac{(z-\mu)^2}{2\sigma^2}} \, dz$$

$$= \sigma^2 \left[ -e^{-\frac{(z-\mu)^2}{2\sigma^2}} \right]_0^\infty + \sqrt{2\pi}\sigma\mu \Pr[Z \ge 0]$$

$$= \sigma^2 e^{-\frac{\mu^2}{2\sigma^2}} + \sqrt{2\pi}\sigma\mu \Pr\left[ \frac{-Z + \mu}{\sigma} \le \frac{\mu}{\sigma} \right]$$

$$= \sqrt{2\pi}\sigma^2 \varphi(x) + \sqrt{2\pi}\sigma\mu\phi(x).$$

*Rearranging yields the desired formula for the first moment. For the second moment, we need to complete the square and perform integration by parts:*

$$\mathbb{E}[ReLU^2(Z)] = \frac{1}{\sqrt{2\pi}\sigma} \int_0^\infty z^2 e^{-\frac{(z-\mu)^2}{2\sigma^2}} \, dz$$

$$= \frac{1}{\sqrt{2\pi}\sigma} \left( \sigma^2 \int_0^\infty (z-\mu) \frac{z-\mu}{\sigma^2} e^{-\frac{(z-\mu)^2}{2\sigma^2}} \, dz + 2\mu \int_0^\infty z e^{-\frac{(z-\mu)^2}{2\sigma^2}} \, dz - \mu^2 \int_0^\infty e^{-\frac{(z-\mu)^2}{2\sigma^2}} \, dz \right)$$

$$= \frac{\sigma^2}{\sqrt{2\pi}\sigma} \left( \left[ -(z-\mu) e^{-\frac{(z-\mu)^2}{2\sigma^2}} \right]_0^\infty + \int_0^\infty e^{-\frac{(z-\mu)^2}{2\sigma^2}} \right) + 2\mu\mathbb{E}[ReLU(Z)] - \mu^2\phi(x)$$

$$= -\sigma\mu\varphi(x) + \sigma^2\phi(x) + 2\mu\mathbb{E}[ReLU(Z)] - \mu^2\phi(x) = \sigma\mu\varphi(x) + (\sigma^2 + \mu^2)\phi(x).$$

**Building Block 3** *Integrals of the form*

$$S^- := \int_{-\infty}^0 z^k \mathcal{N}(z; \mu_1, \sigma_1^2) \mathcal{N}(0; \mu_2, \sigma_2^2) \, dz$$

*where $\mu_1, \mu_2 \in \mathbb{R}, \sigma_1, \sigma_2 > 0$ and $k = 0, 1, 2$ can be efficiently approximated.*

**ESONG PROOF 3** *Employing the substitution $z = -t$ gives*

$$S^- = \mathcal{N}(0; \mu_2, \sigma_2^2) \int_0^\infty (-1)^k t^k \mathcal{N}(-t; \mu_1, \sigma_1^2) \, dt = (-1)^k \mathcal{N}(0; \mu_2, \sigma_2^2) \int_0^\infty t^k \mathcal{N}(t; -\mu_1, \sigma_1^2) \, dt$$

$$= (-1)^k \mathcal{N}(0; \mu_2, \sigma_2^2) \begin{cases} \mathbb{E}[ReLU(\mathcal{N}(-\mu_1, \sigma_1^2))] & \text{for } k = 1, 2 \\ \Pr[-Z \geq 0] = \phi(-\mu_1/\sigma_1) & \text{for } k = 0. \end{cases}$$

Now let $m_{z \to f}(z) = \mathcal{N}(z; \mu_z, \sigma_z^2), m_{a \to f}(a) = \mathcal{N}(a; \mu_a, \sigma_a^2)$ and consider the decomposition

$$m_k = \underbrace{\int_0^\infty z^k \mathcal{N}(z; \mu_z, \sigma_z^2) \mathcal{N}(z; \mu_a, \sigma_a^2) \, dz}_{S^+} + \underbrace{\int_{-\infty}^0 \text{ReLU}^k(z) \mathcal{N}(z; \mu_z, \sigma_z^2) \mathcal{N}(0; \mu_a, \sigma_a^2) \, dz}_{S^-}.$$

Note that $S^+$ falls under Building Block 1 for any $k = 0, 1, 2$. The other addend $S^-$ is equal to 0 for $k = 1, 2$, and is handled by Building Block 3 for $k = 0$.

**Backward Message:** By definition of the Dirac delta, the backward message is equal to

$$m_{f \to z}(z) = \int_{a \in \mathbb{R}} \delta(a - \text{ReLU}(z)) m_{a \to f}(a) \, da = m_{a \to f}(\text{ReLU}(z))$$

which is, of course, not integrable, so it cannot be interpreted as a scaled density. Instead, we apply marginal approximation by deriving formulas for

$$m_k := \int_{z \in \mathbb{R}} z^k m_{z \to f}(z) m_{f \to z}(z) \, dz, \quad k \in \{0, 1, 2\}$$

and then setting

$$m_{f \to z}(z) := \mathcal{N}(z; m_1/m_0, m_2/m_0 - (m_1/m_0)^2) / m_{z \to f}(z).$$

To this end, let $m_{z \to f}(z) = \mathcal{N}(z; \mu_z, \sigma_z^2)$ and $m_{a \to f}(a) = \mathcal{N}(a; \mu_a, \sigma_a^2)$. Then we have

$$m_k = \underbrace{\int_0^\infty z^k \mathcal{N}(z; \mu_z, \sigma_z^2) \mathcal{N}(z; \mu_a, \sigma_a^2) \, dz}_{S^+} + \underbrace{\int_{-\infty}^0 z^k \mathcal{N}(z; \mu_z, \sigma_z^2) \mathcal{N}(0; \mu_a, \sigma_a^2) \, dz}_{S^-}.$$

The two addends $S^+$ and $S^-$ are handled by Building Block 1 and Building Block 3, respectively.

B.2 LEAKY RELU

Another common activation function is the Leaky Rectified Linear Unit

$$\text{LeakyReLU}_\alpha : \mathbb{R} \to \mathbb{R}, z \mapsto \begin{cases} z & \text{for } z \geq 0 \\ \alpha z & \text{for } z < 0. \end{cases}$$

It is parameterized by some $\alpha > 0$ that is typically small, such as $\alpha = 0.1$. In contrast to ReLU, it is injective (and even bijective). For this reason the forward and backward messages are both integrable and can be approximated by both direct and marginal moment matching. The notation is shown in Figure 4.

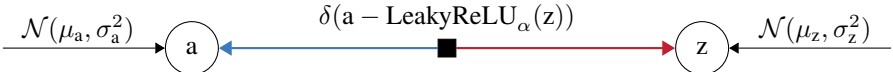

Figure 4: A deterministic factor corresponding to the $\text{LeakyReLU}_\alpha$ activation function.

**Forward Message:** It is easy to show that the density of $\text{LeakyReLU}_\alpha(\mathcal{N}(\mu_z, \sigma_z^2))$ is given by

$$p(a) = \mathcal{N}(\text{LeakyReLU}_{1/\alpha}(a); \mu_z, \sigma_z^2) \begin{cases} 1 & \text{for } z \geq 0 \\ 1/\alpha & \text{for } z < 0 \end{cases}$$

which only has one discontinuity point, namely 0. In particular, it is continuous almost everywhere. So by the density transformation property of Dirac's delta, we have $m_{f \to a}(a) = p(a)$ for almost all $a$. Under the integral we can therefore replace $m_{f \to a}(a)$ by $p(a)$. This justifies that the moments of $m_{f \to a}$ are exactly the moments of $(\text{LeakyReLU}_\alpha)_* \mathcal{N}(\mu_z, \sigma_z^2)$. Its expectation is equal to

$$\mathbb{E}\left[\text{LeakyReLU}_\alpha(\mathcal{N}(\mu_z, \sigma_z^2))\right] = \int_{-\infty}^0 \alpha z \mathcal{N}(z; \mu_z, \sigma_z^2)\, dz + \int_0^\infty z \mathcal{N}(z; \mu_z, \sigma_z^2)\, dz$$

$$= -\alpha \int_0^\infty t \mathcal{N}(t; -\mu_z, \sigma_z^2)\, dt + \int_0^\infty z \mathcal{N}(z; \mu_z, \sigma_z^2)\, dz$$

$$= -\alpha \mathbb{E}[\text{ReLU}(\mathcal{N}(-\mu_z, \sigma_z^2))] + \mathbb{E}[\text{ReLU}(Z)].$$

Both addends are handled by Building Block 2. Yet we can get more insight by further substitution:

$$\mathbb{E}[\text{LeakyReLU}_\alpha(Z)] = -\alpha(\sigma_z \varphi(-\mu_z/\sigma_z) - \mu_z \phi(-\mu_z/\sigma_z)) + \sigma_z \varphi(\mu_z/\sigma_z) + \mu_z \phi(\mu_z/\sigma_z)$$

$$= (1 - \alpha)(\sigma_z \varphi(\mu_z/\sigma_z) + \mu_z \phi(\mu_z/\sigma_z)) + \alpha \mu_z$$

$$= (1 - \alpha)\mathbb{E}[\text{ReLU}(Z)] + \alpha \mathbb{E}[Z].$$

In the second to last equation, we use the identities $\varphi(-x) = \varphi(x)$ and $\phi(-x) = 1 - \phi(x)$. As such, the mean of $\text{LeakyReLU}_\alpha(Z)$ is a convex combination of the mean of $\text{ReLU}(Z)$ and the mean of $Z$. The function $\text{LeakyReLU}_1$ the identity, and its mean is accordingly the mean of $Z$. For $\alpha = 0$, we recover the mean of $\text{ReLU}(Z)$.

The second moment of $\text{LeakyReLU}_\alpha(Z)$ decomposes to

$$\mathbb{E}[\text{LeakyReLU}_\alpha^2(Z)] = \int_{-\infty}^0 \alpha^2 z^2 \mathcal{N}(z; \mu_z, \sigma_z^2)\, dz + \int_0^\infty z^2 \mathcal{N}(z; \mu_z, \sigma_z^2)\, dz$$

$$= \alpha^2 \int_0^\infty z^2 \mathcal{N}(z; -\mu_z, \sigma_z^2)\, dz + \int_0^\infty z^2 \mathcal{N}(z; \mu_z, \sigma_z^2)\, dz$$

$$= \alpha^2 \mathbb{E}[\text{ReLU}^2(\mathcal{N}(-\mu_z, \sigma_z^2))] + \mathbb{E}[\text{ReLU}^2(\mathcal{N}(\mu_z, \sigma_z^2))].$$

Again, both addends are covered by Building Block 2, so approximating the forward message via direct moment matching is feasible.

A marginal approximation can also be found. For all $k = 0, 1, 2$ we have

$$\int_{a \in \mathbb{R}} a^k m_{a \to f}(a) m_{f \to a}(a)\, da = \int_{a \in \mathbb{R}} a^k m_{a \to f}(a) p(a)\, da$$

$$= \underbrace{\frac{1}{\alpha} \int_{-\infty}^0 a^k \mathcal{N}(a; \mu_a, \sigma_a^2) \mathcal{N}(a/\alpha; \mu_z, \sigma_z^2)\, da}_{S^-} + \underbrace{\int_0^\infty a^k \mathcal{N}(a; \mu_a, \sigma_a^2) \mathcal{N}(a; \mu_z, \sigma_z^2)\, da}_{S^+}$$

The term $S^+$ is handled by Building Block 1. The term $S^-$ is equal to

$$S^- = \int_{-\infty}^0 a^k \mathcal{N}(a; \mu_a, \sigma_a^2) \mathcal{N}(a; \alpha \mu_z, (\alpha \sigma_z)^2)\, da$$

$$= (-1)^k \int_0^\infty a^k \mathcal{N}(a; -\mu_a, \sigma_a^2) \mathcal{N}(a; -\alpha \mu_z, (\alpha \sigma_z)^2)\, da$$

and therefore also covered by Building Block 1.

**Backward Message:** By the sifting property of the Dirac delta, the backward message is equal to

$$m_{f \to z}(z) = \int_{a \in \mathbb{R}} \delta(a - \text{LeakyReLU}_\alpha(z)) m_{a \to f}(a) \, da = m_{a \to f}(\text{LeakyReLU}_\alpha(z)).$$

As opposed to ReLU, the backward message is integrable. That means, we can also apply direct moment matching: For all $k = 0, 1, 2$ we have

$$m_{f \to z}(z) = \int_{-\infty}^0 z^k \mathcal{N}(\alpha z; \mu_a, \sigma_a^2) \, dz + \int_0^\infty z^k \mathcal{N}(z; \mu_a, \sigma_a^2) \, dz$$

$$= \frac{(-1)^k}{\alpha} \int_0^\infty z^k \mathcal{N}(z; -\mu_a/\alpha, (\sigma_a/\alpha)^2) \, dz + \int_0^\infty z^k \mathcal{N}(z; \mu_a, \sigma_a^2) \, dz$$

For $k = 1$ or $k = 2$, the integrals fall under Building Block 2 again. If $k = 0$, then

$$m_{f \to z}(z) = \frac{(-1)^k}{\alpha} \phi(-\mu_a/\sigma_a) + \phi(\mu_a/\sigma_a).$$

Again, we can also find a marginal approximation as well. For all $k = 0, 1, 2$, we can write

$$\int_{z \in \mathbb{R}} z^k m_{z \to f}(z) m_{f \to z}(z) \, dz$$

$$= \int_{-\infty}^0 z^k \mathcal{N}(z; \mu_z, \sigma_z^2) \mathcal{N}(\alpha z; \mu_a, \sigma_a^2) \, dz + \int_0^\infty z^k \mathcal{N}(z; \mu_z, \sigma_z^2) \mathcal{N}(z; \mu_a, \sigma_a^2) \, dz$$

$$= \frac{(-1)^k}{\alpha} \int_0^\infty z^k \mathcal{N}(z; -\mu_z, \sigma_z^2) \mathcal{N}(z; -\mu_a/\alpha, (\sigma_a/\alpha)^2) \, dz + \int_0^\infty z^k \mathcal{N}(z; \mu_z, \sigma_z^2) \mathcal{N}(z; \mu_a, \sigma_a^2) \, dz$$

Since both integrals are covered by Building Block 1 we have derived direct and marginal approximations of LeakyReLU messages using moment matching.

### B.3 SOFTMAX

We model the soft(arg)max training signal as depicted in Table 6. For the forward message on the prediction branch, we employ the so-called "probit approximation" (Daxberger et al., 2022):

$$m_{f \to c}(i) = \int \text{softmax}(\mathbf{a})_i \mathcal{N}(\mathbf{a}; \boldsymbol{\mu}, \text{diag}(\boldsymbol{\sigma}^2) \, d\mathbf{a} \approx \text{softmax}(\boldsymbol{t})_i,$$

where $t_j = \mu_j / (1 + \frac{\pi}{8} \sigma_j^2)$, $j = 1, \ldots, d$. For the backward message on a training branch, to say $a_d$, we use marginal approximation. We hence need to compute the moments $m_0, m_1, m_2$ of the marginal of $a_d$ via:

$$m_k = \int a_d^k \text{softmax}(\mathbf{a})_c \mathcal{N}(\mathbf{a}; \boldsymbol{\mu}, \text{diag}(\boldsymbol{\sigma}^2)) \, d\mathbf{a}$$

$$= \int_{a_d} a_d^k \mathcal{N}(a_d; \mu_d, \sigma_d^2) \int_{\mathbf{a} \backslash a_d} \text{softmax}(\mathbf{a})_i \prod_{j \neq i} \mathcal{N}(a_j; \mu_j, \sigma_j^2) \, d(\mathbf{a} \backslash a_d) da_d.$$

We can reduce the inner integral to the probit approximation by regarding the point distribution $\delta_{a_d}$ as the limit of a Gaussian with vanishing variance:

$$\int_{\mathbf{a} \backslash a_d} \text{softmax}(\mathbf{a})_c \prod_{j \neq d} \mathcal{N}(a_j; \mu_j, \sigma_j^2) \, d(\mathbf{a} \backslash a_d)$$

$$= \int_{\mathbf{a} \backslash a_d} \int_{\tilde{a}_d} \delta(\tilde{a}_d - a_d) \text{softmax}(a_1, \ldots, a_{d-1}, \tilde{a}_d) \prod_{j \neq d} \mathcal{N}(a_j; \mu_j, \sigma_j^2) \, d\tilde{a}_d \, d(\mathbf{a} \backslash a_d)$$

$$= \int_{\tilde{\mathbf{a}} \backslash \tilde{a}_d} \lim_{\sigma \to 0} \int_{\tilde{a}_i} \text{softmax}(\tilde{\mathbf{a}})_c \mathcal{N}(\tilde{a}_d; a_d, \sigma^2) \prod_{j \neq d} \mathcal{N}(\tilde{a}_j; \mu_j, \sigma_j^2) \, d\tilde{a}_i \, d\tilde{\mathbf{a}}_i$$

By Lebesgue's dominated convergence theorem we obtain equality to

$$\lim_{\sigma \to 0} \int_{\tilde{\mathbf{a}}} \mathrm{softmax}(\tilde{\mathbf{a}})_c \mathcal{N}(\tilde{a}_d; a_d, \sigma^2) \prod_{j \neq i} \mathcal{N}(\tilde{a}_j; \mu_j, \sigma_j^2) \, d\tilde{\mathbf{a}}$$

$$\approx \lim_{\sigma \to 0} \mathrm{softmax}(\boldsymbol{t})_i = \mathrm{softmax}(t_1, \ldots, t_{d-1}, a_d) \quad \text{where} \quad t_j = \begin{cases} \mu_j/(1 + \frac{\pi}{8}\sigma_j^2) & \text{for } j \neq d \\ a_d/(1 + \frac{\pi}{8}\sigma^2) & \text{for } j = d. \end{cases}$$

Hence, we can approximate $m_k$ by one-dimensional numerical integration of

$$m_k \approx \int_{a_d} a_d^k \mathcal{N}(a_d; \mu_d, \sigma_d^2) \, \mathrm{softmax}(t_1, \ldots, t_{d-1}, a_d) \, da_d.$$

## C  EXPERIMENTAL SETUP

**Synthetic Data - Depth Scaling:**  We generated a dataset of 200 points by randomly sampling $x$ values from the range $[0, 2]$. The true data-generating function was

$$f(x) = 0.5x + 0.2 \sin(2\pi \cdot x) + 0.3 \sin(4\pi \cdot x).$$

The corresponding $y$ values were sampled by adding Gaussian noise: $f(x) + \mathcal{N}(0, 0.05^2)$. For the architecture, we used a three-layer NN with the structure:

$$[\mathrm{Linear}(1, 16), \mathrm{LeakyReLU}(0.1), \mathrm{Linear}(16, 16), \mathrm{LeakyReLU}(0.1), \mathrm{Linear}(16, 1)].$$

A four-layer network has one additional $[\mathrm{Linear}(16, 16), \mathrm{LeakyReLU}(0.1)]$ block in the middle, and a five-layer network has two additional blocks. For the regression noise hyperparameter, we used the true noise $\beta^2 = 0.05^2$. The models were trained for 500 iterations over one batch (as all data was processed in a single active batch).

**Synthetic Data - Uncertainty Evaluation:**  The same data-generation process was used as in the depth-scaling experiment, but this time, $x$ values were drawn from the range $[-0.5, 0.5]$. The network architecture remained the same as the three-layer network, but the width of the layers was increased to 32. We trained 100 networks with different random seeds on the same dataset. We define a $p$-credible interval for $0 \leq p \leq 1$ as:

$$[\mathrm{cdf}^{-1}(0.5 - \frac{p}{2}), \mathrm{cdf}^{-1}(0.5 + \frac{p}{2})].$$

For each credible interval mass $p$ (ranging from 0 to 1 in steps of 0.01), we measured how many of the $p$-credible intervals (across the 100 posterior approximations) covered the true data-generating function. This evaluation was done at each possible $x$ value (ranging from -20 to 20 in steps of 0.05), generating a coverage rate for each combination of $p$ and $x$. For each $p$, we then computed the median for $x > 10$ and the median for $x < -10$. If we correlate the $p$ values with the medians, we found that for the median obtained from positive $x$ values the correlation was 0.96, for negative $x$ it was 0.99, and for the combined set of medians it was 0.9.

**CIFAR-10:**  For our CIFAR-10 experiments, we used the default train-test split and trained the following feed-forward network:

```
class Net(nn.Module):
    def __init__(self):
        super(Net, self).__init__()
        self.model = nn.Sequential(
            # Block 1
            nn.Conv2d(3, 32, 3, padding=0),
            nn.LeakyReLU(0.1),
            nn.Conv2d(32, 32, 3, padding=0),
            nn.LeakyReLU(0.1),
            nn.MaxPool2d(2),
            # Block 2
            nn.Conv2d(32, 64, 3, padding=0),
            nn.LeakyReLU(0.1),
```

```
                nn.Conv2d(64, 64, 3, padding=0),
                nn.LeakyReLU(0.1),
                nn.MaxPool2d(2),
                # Head
                nn.Flatten(),
                nn.Linear(64 * 5 * 5, 512),
                nn.LeakyReLU(0.1),
                nn.Linear(512, 10),
            )

        def forward(self, x):
            return self.model(x)
```

In the case of AdamW and IVON we trained with a cross-entropy loss on the softargmax of the network output. For our message passing method we used our argmax factor as a training signal instead of softargmax, see Appendix F. The reason is that for softargmax we only have message approximations relying on rather expensive numerical integration. In our library this factor graph can be constructed via

```
    fg = create_factor_graph([
                size(d.X_train)[1:end-1], # (3, 32, 32)
                # First Block
                (:Conv, 32, 3, 0), # (32, 30, 30)
                (:LeakyReLU, 0.1),
                (:Conv, 32, 3, 0), # (32, 28, 28)
                (:LeakyReLU, 0.1),
                (:MaxPool, 2), # (32, 14, 14)
                # Second Block
                (:Conv, 64, 3, 0), # (64, 12, 12)
                (:LeakyReLU, 0.1),
                (:Conv, 64, 3, 0), # (64, 10, 10)
                (:LeakyReLU, 0.1),
                (:MaxPool, 2), # (64, 5, 5)
                # Head
                (:Flatten,), # (64*5*5 = 1600)
                (:Linear, 512), # (512)
                (:LeakyReLU, 0.1),
                (:Linear, 10), # (10)
                (:Argmax, true)
            ], batch_size)
```

For all methods we used a batch size of 128 and trained for 25 epochs with a cosine annealing learning rate schedule. Concerning hyperparameters: For AdamW we found the standard parameters of $lr = 10^{-3}, \beta_1 = 0.9, \beta_2 = 0.999, \epsilon = 10^{-8}$ and $\delta = 10^{-4}$ to work best. For IVON we followed the practical guidelines given in the Appendix of Shen et al. (2024).

To measure calibration, we used 20 bins that were split to minimize within-bin variance. For OOD recognition, we predicted the class of the test examples in CIFAR-10 (in-distribution) and SVHN (OOD) and computed the entropy over softmax probabilities for each example. We then sort them by negative entropy and test the true positive and false positive rates for each possible (binary) decision threshold. The area under this ROC curve is computed in the same way as for relative calibration.

## D  PRIOR ANALYSIS

The strength of the prior determines the amount of data needed to obtain a useful posterior that fits the data. Our goal is to draw prior means and set prior variances so that the computed variances of all messages are on the order of $\mathcal{O}(1)$ regardless of network width and depth. It is not entirely clear if this would be a desirable property; after all, adding more layers also makes the network more expressive and more easily able to model functions with very high or low values. However, if we

let the predictive prior grow unrestricted, it will grow exponentially, leading to numerical issues. In the following, we analyze the predictive prior under simplifying assumptions to derive a prior initialization that avoids exponential variance explosion. While we fail to achieve this goal, our current prior variances are still informed by this analysis.

In the following, we assume that the network inputs are random variables. Then, the parameters of messages also become random variables, as they are derived from the inputs according to the message equations. Our goal is to keep the expected value of the variance parameter of the outgoing message at a constant size. We also assume that the means of the prior are sampled according to spectral initialization, as described in Section 4.3.

FIRSTGAUSSIANLINEARLAYER - INPUT IS A CONSTANT

Each linear layer transforms some $d_1$-dimensional input $\mathbf{x}$ to some $d_2$-dimensional output $\mathbf{y}$ according to $\mathbf{y} = W\mathbf{x} + \mathbf{b}$. In the first layer, $\mathbf{x}$ is the input data. For this analysis, we assume each element $x_i$ to be drawn independently from $x_i \sim \mathcal{N}(0, 1)$. Let $\mathbf{x}$ be a $d_1$-dimensional input vector, $\mathbf{m}_w$ be the prior messages from one column of $W$, and $z = \mathbf{w}'\mathbf{x}$ be the vector product before adding the bias.

During initialization of the weight prior, we draw the prior means using spectral parametrization and set the prior variances to a constant:

$$m_{w_i} = \mathcal{N}(\mu_{w_i}, \sigma_w^2) \text{ with } \mu_{w_i} \sim \mathcal{N}(0, l^2),$$

$$l = \frac{1}{\sqrt{k}} \cdot \min(1, \sqrt{\frac{d_2}{d_1}}).$$

By applying the message equations, we then approximate the forward message to the output with a normal distribution

$$m_z = \mathcal{N}(\mu_z, \sigma_z^2).$$

Because $\sigma_z^2$ depends on the random variables $x_i$, it is also a random variable that follows a scaled chi-squared distribution

$$\sigma_z^2 = \sum_{i=1}^{d_1} x_i^2 \cdot \sigma_w^2$$

$$\sigma_z^2 \sim \chi_{d_1}^2 \cdot \sigma_w^2$$

and its expected value is

$$\mathbb{E}[\sigma_z^2] = d_1 \cdot \sigma_w^2.$$

We conclude that we can control the magnitude of the variance parameter by choosing $\mathbb{E}[\sigma_z^2]$ and setting $\sigma_w^2 = \frac{\mathbb{E}[\sigma_z^2]}{d_1}$.

GAUSSIANLINEARLAYER - INPUT IS A VARIABLE

In subsequent linear layers, the input $\mathbf{x}$ is not observed and we receive an approximate forward message that consists of independent normal distributions

$$m_{x_i} = \mathcal{N}(\mu_{x_i}, \sigma_{x_i}^2).$$

Following the message equations, the outgoing forward message to $z$ then has a variance

$$\sigma_z^2 = \sum_{i=1}^{d_1} (\sigma_{x_i}^2 + \mu_{x_i}^2) \cdot (\sigma_w^2 + \mu_{w_i}^2) - (\mu_{x_i}^2 * \mu_{w_i}^2)$$

$$= \sum_{i=1}^{d_1} \underbrace{\sigma_{x_i}^2 \cdot \sigma_w^2}_{\text{I}} + \underbrace{\sigma_{x_i}^2 \cdot \mu_{w_i}^2}_{\text{II}} + \underbrace{\mu_{x_i}^2 \cdot \sigma_w^2}_{\text{III}}$$

The layer's prior variance $\sigma_w^2$ is a constant, whereas all other elements are random variables according to our assumptions. To make further analysis tractable, we also have to assume that the variances $\sigma_{x_i}^2$ of the incoming forward messages are identical constants for all $i$, not random variables. We furthermore assume that the means are drawn i.i.d. from:

$$\mu_{w_i} \sim \mathcal{N}(0, l^2)$$
$$\mu_{x_i} \sim \mathcal{N}(\mu_{\mu_x}, \sigma_{\mu_x}^2).$$

The random variable $\sigma_z^2$ then follows a generalized chi-squared distribution

$$\sigma_z^2 \sim \left( \sum_{i=1}^{d_1} \underbrace{\sigma_x^2 \cdot l^2 \cdot \chi^2(1, 0^2)}_{\text{II}} + \underbrace{\sigma_w^2 \cdot \sigma_{\mu_x}^2 \cdot \chi^2(1, \mu_{\mu_x}^2)}_{\text{III}} \right) + \underbrace{d_1 \cdot \sigma_w^2 \cdot \sigma_x^2}_{\text{I}}$$

and its expected value is

$$\mathbb{E}[\sigma_z^2] = \left( \sum_{i=1}^{d_1} \sigma_x^2 \cdot l^2 \cdot (1 + 0^2) + \sigma_w^2 \cdot \sigma_{\mu_x}^2 \cdot (1 + \mu_{\mu_x}^2) \right) + d_1 \cdot \sigma_w^2 \cdot \sigma_x^2$$

$$= d_1 \cdot \left( \sigma_x^2 \cdot l^2 + \sigma_w^2 \cdot \sigma_{\mu_x}^2 \cdot (1 + \mu_{\mu_x}^2) + \sigma_w^2 \cdot \sigma_x^2 \right)$$

$$= \underbrace{d_1 \cdot \sigma_x^2 \cdot l^2}_{\text{II}} + \underbrace{d_1 \cdot (\sigma_{\mu_x}^2 \cdot (1 + \mu_{\mu_x}^2) + \sigma_x^2) \cdot \sigma_w^2}_{\text{I+III}}.$$

As $\sigma_w^2$ has to be positive, we conclude that if we choose $\mathbb{E}[\sigma_z^2] > d_1 \cdot \sigma_x^2 \cdot l^2$, then we can set

$$\sigma_w^2 = \frac{\mathbb{E}[\sigma_z^2] - d_1 \cdot \sigma_x^2 \cdot l^2}{d_1 \cdot (\sigma_{\mu_x}^2 \cdot (1 + \mu_{\mu_x}^2) + \sigma_x^2)}.$$

We know (or choose) $d_1$, $l^2$, and $\mathbb{E}[\sigma_z^2]$, but we require values for $\sigma_x^2$, $\mu_{\mu_x}^2$, and $\sigma_{\mu_x}^2$ to be able to choose $\sigma_w^2$. We will find empirical values for these parameters in the next section.

### EMPIRICAL PARAMETERS + LEAKYRELU

To inform the choice of the prior variances of the inner linear layers, we also need to analyze LeakyReLU. We assume the network is an MLP that alternates between linear layers and LeakyReLU. As the message equations of LeakyReLU are too complicated for analysis, we instead use empirical approximation. Let $m_a = \mathcal{N}(\mu_a, \sigma_a^2)$ be an incoming message (from the pre-activation variable to LeakyReLU). We assume that $\sigma_a^2 = t$ is a constant and that $\mu_a \sim \mathcal{N}(0, 1)$ is a random variable. By sampling multiple means and then computing the outgoing messages (after applying LeakyReLU), we can approximate the average variance of the outgoing messages, as well as the average and empirical variance over means of the outgoing messages.

We computed these statistics for 101 different leak settings with 100 million samples each, and found that the relationship between leak and $\mu_{\mu_x}$ (average mean of the outgoing message) is approximately linear, while the relationships between leak and $\sigma_{\mu_x}^2$ or $\mu_{\sigma_x^2}$ are approximately quadratic. Using these samples, we fitted coefficients with an error margin below $5 \cdot 10^{-5}$. For our network, we chose a target variance of $1.5$ and a leak of $0.1$, resulting in

$$\sigma_x^2 = 0.8040586726631379$$
$$\sigma_{\mu_x}^2 \cdot (1 + \mu_{\mu_x}^2) = 0.44958619556324186.$$

These values are sufficient for now setting the prior variances of the inner linear layer according to the equations above. Finally, we set the prior variance of the biases to $0.5$, so that the output of each linear layer achieves an overall target prior predictive variance of approximately $t = 1.5 + 0.5 = 2.0$.

RESULTS IN PRACTICE

In practice, we found that the variance of the predictive posterior still goes up exponentially with the depth of the network despite our derived prior choices. However, if we lower the prior variance further to avoid this explosion, the network is overly restricted and unable to obtain a good fit during training. We therefore set the prior variances as outlined here, but acknowledge that choosing a good prior is a general known problem.

# E    EVALUATION: ANALYSIS ON TABULAR DATA

## E.1    GOAL, SETUP & METHODS

In thefollowing, we benchmark the proposed Bayesian neural network (BNN) approach using approximate message passing on a suite of classical regression tasks yet. It is crucial to understand the behavior of BNNs on classic regression tasks of medium difficulty. Our goal is to assess the strengths and weaknesses of BNNs, especially regarding performance and overfitting behavior.

We will analyze if our Bayesian neural networks keep their promise of delivering a good calibration, in particular if the estimated uncertainty matches the errors we are seeing. We are using a suite of regression problems mainly from the widespread UCI machine learning repository Dua & Graff (2017). This repository contains hundreds of publicly available datasets that are used by researchers as standard benchmarks to test new algorithm approaches. We focus on regression tasks with up to 20 features. Here are the datasets used in our analysis:

- California Housing: The California Housing dataset comprises 8 numerical features derived from the 1990 U.S. Census data. The aim is to estimate the median house value in a specific area, based on nine features with information about the neighborhood. These features include median income, median house age, total number of rooms, total number of bedrooms, population, number of households, latitude, longitude, and a categorical variable for ocean proximity. It has the options "near bay" (San Francisco Bay), "near ocean", "less than one hour to the ocean", and "inland". Challenges in modeling this dataset involve capturing non-linear relationships and spatial dependencies, as well as influencing factors that are not included in the feature set. Nugent (2017)

- Abalone: The Abalone dataset is about predicting the age of these specific snakes by measurements. It contains 4,177 instances with 8 input features: one categorical feature (sex) and seven continuous features (length, diameter, height, whole weight, shucked weight, viscera weight, and shell weight). The target variable is the number of rings, which correlates with the age of the abalone. A significant challenge is the non-linear relationship between the physical measurements and age, as well as the presence of outliers and multicollinearity among features. Nash et al. (1994)

- Wine Quality: This dataset includes two subsets related to red and white "Vinho Verde" wines from Portugal, each with 11 physicochemical input variables such as fixed acidity, volatile acidity, citric acid, residual sugar, chlorides, free sulfur dioxide, total sulfur dioxide, density, pH, sulphates, and alcohol. All of these variables are continuous. Furthermore, there is one binary variable indicating if the sample is a white or red whine. The target variable is the wine quality score (0–10) rated by wine tasters. Challenges include class imbalance, as most wines have medium quality scores, and the subjective nature of the quality ratings. Cortez et al. (2009)

- Bike Sharing: The Bike Sharing dataset is about predicting the usage of rental bikes in an area based on seasonal information, weather, and usage profiles. Specifically, it contains contains hourly and daily counts of rental bikes in the Capital Bikeshare system from 2011 to 2012, along with 12 features including season, year, month, day, weekday, hour, holiday, working day, weather situation, temperature, "feels like" temperature, humidity, wind speed, number of casual users, and number of registered users. The target variable is the count of total rental bikes. Modeling challenges involve capturing complex temporal patterns, handling missing data, and accounting for external factors like weather and holidays. Fanaee-T (2013)

- Forest Fires: This dataset comprises 517 instances with 12 features: spatial coordinates (X, Y), temporal variables (month, day), and meteorological data (FFMC, DMC, DC, ISI, temperature, relative humidity, wind, and rain). The target is to predict the burned area of the forest (in hectares) in the northeast region of Portugal in wild fires. The primary challenge is the high skewness of the target variable, with many instances having a burned area of zero, making it difficult to model and evaluate performance accurately. Cortez & Morais (2007)

- Heart Failure: The Heart Failure Clinical Records dataset includes 299 patient records with 13 clinical features such as age, anaemia, high blood pressure, creatinine phosphokinase, diabetes, ejection fraction, platelets, serum creatinine, serum sodium, sex, smoking, time, and death event. The target variable is a binary indicator of death occurrence. Challenges include the small dataset size, potential class imbalance, and missing values, which can affect the generalizability of predictive models. hea (2020)

- Real Estate Taiwan: This dataset is about predicting house prices in New Taipei City, Taiwan. It contains 414 instances with 6 features: transaction date, house age, distance to the nearest MRT station, number of convenience stores, latitude, and longitude. The target variable is the house price per unit area. Challenges in modeling this dataset involve capturing the influence of location-based features and dealing with a low sample size. Yeh (2018)

Table 3: Summary of Regression Datasets

| Dataset | Samples | Features | Feature Types |
|---|---|---|---|
| California Housing | 20,640 | 9 | Numerical: latitude, longitude, house median age, total rooms, total bedrooms, population, households, median income; categorical: ocean proximity |
| Abalone | 4,177 | 8 | 1 categorical (sex), 7 numerical: length, diameter, height, whole weight, shucked weight, viscera weight, shell weight. |
| Wine Quality (Red) | 1,599 | 11 | All numerical: fixed acidity, volatile acidity, citric acid, residual sugar, chlorides, free sulfur dioxide, total sulfur dioxide, density, pH, sulphates, alcohol. |
| Wine Quality (White) | 4,898 | 11 | Same as red wine dataset. |
| Bike Sharing | 17,379 | 12 | Mix of categorical and numerical: season, year, month, hour, holiday, weekday, working day, weather situation, temperature, feels-like temperature, humidity, wind speed. |
| Forest Fires | 517 | 12 | 2 categorical (month, day), 10 numerical: FFMC, DMC, DC, ISI, temperature, relative humidity, wind, rain, X, Y coordinates. |
| Heart Failure | 299 | 13 | Mix of binary and numerical: age, anaemia, high blood pressure, creatinine phosphokinase, diabetes, ejection fraction, platelets, serum creatinine, serum sodium, sex, smoking, time, death event. |
| Real Estate Taiwan | 414 | 6 | All numerical: transaction date, house age, distance to nearest MRT station, number of convenience stores, latitude, longitude. |

For a quick comparison, the number of features and samples are shown in Table 3. As we can see, the datasets have very different sizes. This is an important challenge where we want to evaluate the Bayesian neural network.

For our training, we normalize the values in all columns. In particular, for each column $c$, we subtract the mean of $c$ from each entry, and divide by the empiric standard deviation. This applies to both feature and target columns. The primary reason for us to do this is the fact that the BNN is designed to handle input with a mean of zero and a standard deviation of one best. Standard neural networks perform best with numbers in this range as well. Furthermore, normalization helps with the common

problem that different columns have values in different orders of magnitude before normalization, for example tens of thousands for yearly income, and small numbers for number of bathrooms in the case of California housing. An additional advantage is that performance can be compared across datasets (approximately), highlighting strengths and weaknesses across different settings.

On these datasets, we apply a random split into training and test dataset, where we dedicate 80% on the training dataset and 20% on the test dataset. Then, we run a pipeline where we evaluate the performance on for each dataset both on the Bayesian neural network implemented in Julia, and on the standard neural network implemented in PyTorch.

The Julia implementation uses a neural network with two hidden layers, and 64 neurons per layer. LeakyReLy is the activation function, and the default standard deviation at the last layer is set to $0.4$.

The PyTorch network uses two hidden layers with 64 neurons per hidden layer as well. The only difference in terms of architecture to the BNN is the fact that stardard ReLu is used. We train with the same learning rate of $3 \cdot 10^{-3}$. When training with these static datasets which are relatively small, overfitting is a huge challenge. It is especially the challenge of overfitting, that should be addressed by our Bayesian neural networks. As the results will show, overfitting is a significant problem on these datasets for the PyTorch network. To tackle overfitting, several regularization techniques have been proposed. However, overfitting remains a fundamental flaw of the classical neural networks. To aim for a fair comparison, we add a weight-decay regularization to the default setup of the PyTorch networks. The specific weight decay parameter is $1 \cdot 10^{-4}$.

For both candidates, we use a batch size of 256. Training is done over a horizon of 500 epochs which corresponds to very different training lengths, due to the different sizes of the datasets. Hyperparameter tuning was done for none of the datasets.

The primary metric to rate performance on our regression datasets is the root mean squared error between the labels and the predicted values. Because we have a standardized output, trivial benchmarks like the constant zero function give an RMSE of one. Hence, we expect RMSE values from the model to substantially improve over one.

## E.2 RESULTS

When running the training an evaluation pipeline, we measure the RMSE of the train and validation dataset after each batch. That naturally comes with small fluctuations, and a slightly uneven-looking learning curve. On the larger datasets, that implies a substantially lower variance of the loss estimation for the validation dataset, and a much smoother learning curve. On the smaller datasets, the evaluation set is quite small, and therefore, the variance is high.

| Dataset | BNN Train | BNN Val | PyTorch Train | PyTorch Val | BNN Val / PyTorch Val (%) |
|---|---|---|---|---|---|
| Abalone | 0.5748 | 0.5965 | 0.4643 | 0.6668 | 89.46% |
| Wine Quality | 0.7682 | 0.8095 | 0.3198 | 0.7722 | 104.83% |
| California Housing | 0.5455 | 0.5764 | 0.3427 | 0.4278 | 134.73% |
| Bike Sharing | 0.2219 | 0.2432 | 0.1557 | 0.216 | 112.58% |
| Forest Fires | 1.0537 | 1.2015 | 0.0302 | 0.4113 | 292.14% |
| Heart Failure | 0.9957 | 0.9605 | 0.0034 | 0.9107 | 105.47% |
| Real Estate Taiwan | 0.6229 | 0.5823 | 0.1438 | 0.5431 | 107.22% |

Table 4: Comparison of minimum RMSE for BNN (Julia) and PyTorch approaches. The data was obtained by running the respective training scripts for 500 epochs and measuring the root mean squared error on training and validation splits.

In Table 4, the best performances on the datasets along a training run are reported. For each of the metrics, we are taking the minimum value over all trained batches. Due to the variance of these estimations, the minimum underestimates the true minimal loss. But the calculation and batch sizes are the the same for both setups, training and loss, so it does not affect the viability of the comparison. Note that the minimum for train and test are not necessarily obtained at the same step.

As the comparison column in Table 4 show, the minimal root mean squared errors are similar for many datasets. On *wine quality*, and *California housing*, *bike sharing*, and *real estate Taiwan*, the PyTorch neural network has advantages in terms of pure regression performance, but only for California housing this difference is considered strong. On the other datasets, PyTorch's performance advantage is only slight, and within the range of hyperparameter tuning. As we will see in the later analysis, the BNN has not finished to learn after 500 epochs. Our experiment for 5,000 epochs reveils that the performance difference actually shrinks to *x* percent. On Abalone, the BNN's pure performance is even superior. For *forest fires*, we see that the Bayesian neural network was not able to solve the regression problem in this setup (RMSE is larger than 1), but PyTorch achieved respectable predictive power. Neither of the two models was capable to actually solve the *heart failure* dataset, where both models gave only slightly better RMSE performance than one. The detailed learning curves for these four datasets with similar performance are plotted in Figure 5.

A general trend that is visible both in Table 4 and in Figure 5 is the fact BNNs show a minimal level of overfitting, even without further regularization. Depending on the dataset, PyTorch shows high or very high levels of overfitting, even though regularization was applied.

As one would expect, PyTorch's overfitting problem is the least serious on California housing and bike sharing (Figure 5a and 5d), the datasets with the most available samples (California housing has 16,512 training samples, and bike sharing has 13,903 training samples). Over the course of the training, the root mean squared error of the validation and train dataset decrease together, but at around 1/5 of the entire training run, the validation loss stalls, and the training loss decreases further – the model is just overfitting.

The BNN's training on these datasets stays above PyTorch's learning curves after the rapid first initial improvement, and slowly improves towards PyTorch's level. While it matches PyTorch's performance on bike sharing after 500 epochs, it still has a performance disadvantage on California Housing of 34%. Nevertheless, with additional training, this performance difference shrinks down to only x percent after 5,000 epochs, a very respectable performance. However, the most notable feature on these two training runs is the observation that the BNN only shows minimal overfitting. The training and validation curve decrease on-par, the existing and expected slight overfitting does not increase by time. The minimal train RMSE is 0.546 compared to a validation RMSE of 0.576 on California housing in the first 500 epochs, only 5%! On PyTorch, we see 20% overfitting with increasing trend. On bike sharing, the BNN overfits by 9% compared to 28% for the PyTorch network. The fact that overfitting is not more severe on these datasets is thanks to the large number of samples, around three to four times more than the number of parameters of the model.[7]

Wine quality and abalone both have roughly the same sample size as number of parameters. Therefore, we would expect that overfitting becomes a much more severe issue for these datasets. As Figure 5b and 5c show, the overfitting problem on these two datasets has increased to a severe level. In both cases, the overfitting starts right from the beginning, with a steady improvement of the train loss, but a validation loss that soon finds its minimum, and increases again. In the case of wine quality, this behavior is significant, but stable and smooth. On abalone however, the validation loss behaves very unstable, and never comes close to the BNN's performance.

The BNN gets close to its optimal performance very rapidly in both cases. The overall training is much faster than for California housing. Again, we see minimal levels of overfitting, a behavior that is very similar to the analysis of the previous two datasets. Moreover, the existing overfitting does not increase with time but stays at a similar level throughout the training horizon of 500 epochs.

The last three datasets, forest fires, heart failure and real estate Taiwan, are quite important, and also difficult to analyze. They only have a few hundred data points, and tend to be very imbalanced. On the UCI repo and in the literature, they are marked as very difficult datasets. For example, most

---

[7]The models have two fully connected hidden layers, one transition to the first layer, and the transition to the regression output. Hence, the models have $64 + 64^2 + 64 = 4224$ weights.

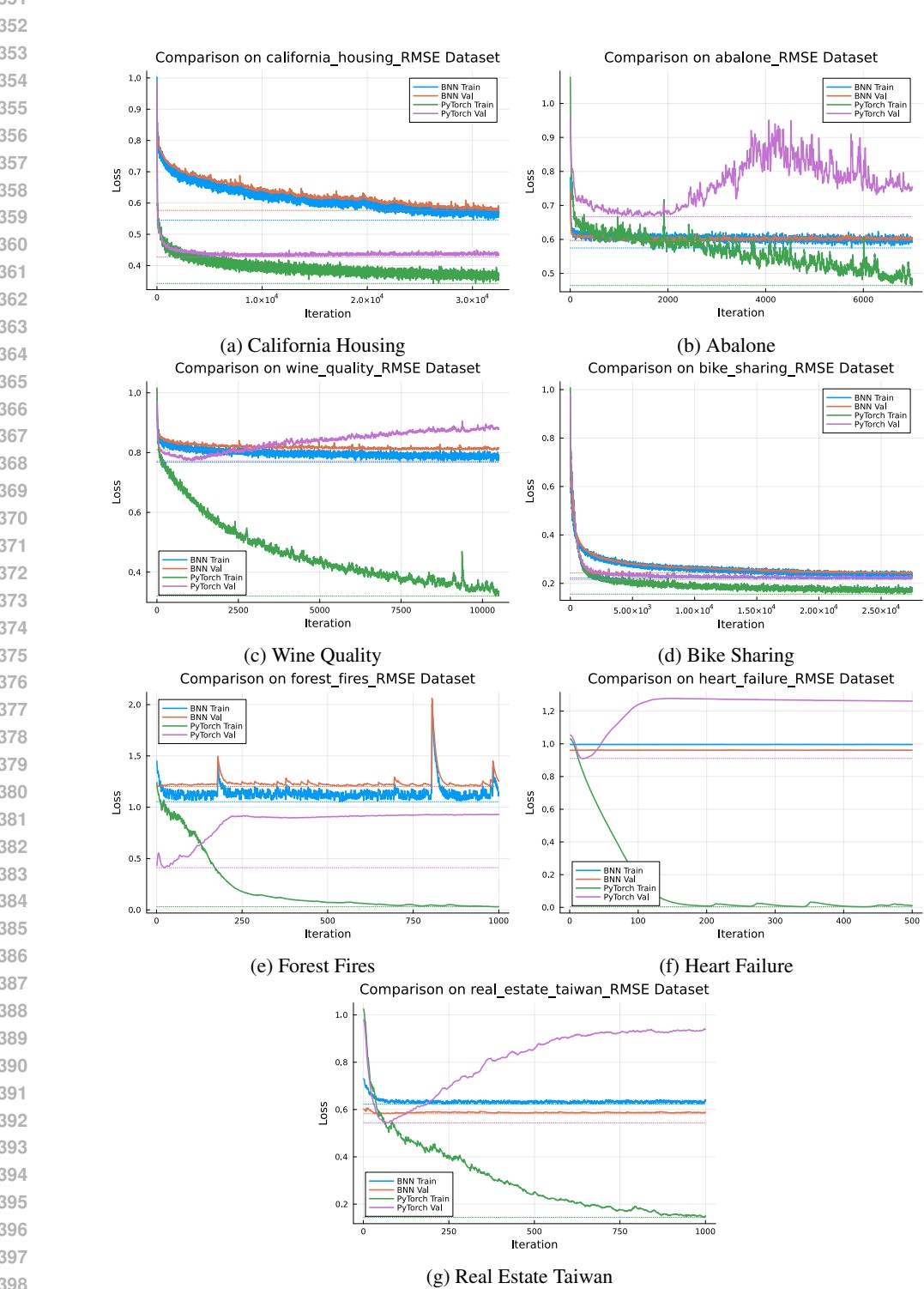

(a) California Housing

(b) Abalone

(c) Wine Quality

(d) Bike Sharing

(e) Forest Fires

(f) Heart Failure

(g) Real Estate Taiwan

Figure 5: Learning curves of approximate message-passing Bayesian neural networks against PyTorch neural networks

samples in forest fires contain a fire with zero acres burned, and only very few samples with a high amount of burned wood. Additionally, the validation datasets get so small, that biases could easily find their way into the validation dataset. The concrete performance numbers must therefore be treated with care, and the strange effect of smaller validation loss than train loss can be observed for both PyTorch and BNN on some of the runs. For example, it seems very likely that PyTorch's good performance on forest fires was obtained randomly by lucky parameter initialization instead of actual performance. Hence, based on Figure 5e, Figure 5f, and Table 4, we consider the forest fire dataset and the heart failure datasets unsolved by both models. As we would expect for a model with more than 10x more parameter than training samples, the train loss goes to zero for the PyTorch model. The Bayesian neural network is unable to learn the datasets as well, but it does not overfit. Its training loss never creates the impression that the loss was any lower than it actually is.

The dataset on estimating Taiwanese real estate (learning curve Figure 5g) is the only of the small datasets that actually gets solved to an acceptable level by both models. Again, we see very heavy overfitting by the PyTorch model, and no overfitting by the BNN. (Actually, this is one of the cases where the validation loss is lower than the train loss, probably due to a biased train/test split.) Moreover, the BNN achieves this performance after only a few epochs.

From this analysis, we can note the following learnings:

1. Our approximate message-passing Bayesian neural networks can achieve similar performance like PyTorch neural networks with the same architecture. In most cases, they stay slightly behind in terms of raw performance, but sometimes outperform the standard implementation.

2. Our Bayesian neural networks apparently do not share the fundamental flaw of overfitting. Their train loss often is slightly lower than the validation loss, as it is expected from any type of machine learning model, but the train and validation loss do not detach and differences in these two curves remain low.

3. Bayesian neural networks learn fast on small datasets, and learn slower on larger datasets like California housing.

The last point is worth some deeper investigation. Why does the model learn the Taiwanese real estate after a few epochs, and requires many more epochs for California housing, although one single epoch is already 40 times larger than on the Taiwanese real estate dataset? Bayesian model's learning gets significantly slower over time. While the learning rate remains constant on the standard PyTorch implementation, the speed of change of the weights in the BNN decreases with the variances of the weights. And the variances decrease once more data has been learned. Therefore, a sample at the end of a large dataset has less power to change the model's parameters, and learning slows down. As a conclusion, the BNNs are especially valuable when only few training samples are available, and when overfitting should be avoided.

### E.3 CAN THE BNNS ESTIMATE THEIR OWN UNCERTAINTY?

The absence of overfitting on BNNs is a side product of the metric that Bayesian methods traditionally try to optimize, the calibration. In contrast to classic neural networks, our BNNs output their mean $\mu(x)$ together with an estimated standard deviation $\sigma(x)$ expressing the uncertainty for an input $x$. We can use the ground truth $y_x$ to analyze how well the uncertainty was estimated. Because the ground-truth values were normalized during pre-processing, we expect the $\mu$s to also have a mean close to zero, with a variance of roughly one.[8] Specifically, we can calculate the z-score of the observation as

$$z(x) = \frac{\mu(x) - y_x}{\sigma(x)}. \tag{7}$$

If the model was perfectly calibrated, these z-scores would follow a perfect standard normal distribution. That does not mean that the model perfectly predicts the ground truth, and it also does not mean that the uncertainty estimation is correct every single time. Instead, it means that the errors follow the same distribution as predicted by the model. As a rule of thumb, in 68% of the cases, $y_x$ should be

---

[8] Of course, the empiric variance of these $\mu$s should not be mixed up with $\sigma(x)$, which expresses a completely different concept, and it is usually much smaller than zero.

within $[\mu(x) - \sigma(x), \mu(x) + \sigma(x)]$, in 95% of the cases within $[\mu(x) - 2\sigma(x), \mu(x) + 2\sigma(x)]$, and in 99% within $[\mu(x) - 3\sigma(x), \mu(x) + 3\sigma(x)]$.

In our experiments, we take the model after 500 epochs for each of the datasets, and calculate the z-scores. We can blindly take the last model because we do not see decreasing performance or problematic overfitting for the BNNs. Then, we use kernel density estimation with Gaussian kernels to obtain an empirical error distribution that is compared to the standard normal distribution. To systematically compare the two distributions, we could report the approximate KL divergence. However, few of our readers have an intuitive understanding of what specific KL numbers mean, and we do not have a comparison partner. Hence, we illustrate the distributions in Figure 6 to best communicate the calibration of the BNN.

As we can see, the calibration is quite good on most of these datasets. When it comes to California housing (Figure 6a), and abalone (Figure 6b), the uncertainty of the model matches the errors quite well. On wine quality (Figure 6c), the model underestimates the errors, but the uncertainty estimation is still usable. The opposite is true for the bike sharing dataset. As previously discussed, the model solves this dataset very well, and the predicted uncertainties underestimate the true errors (Figure 6d). The wine quality dataset and the bike sharing dataset cannot be solved by the BNN. Hence, the error distributions are rather weak (Figure 6e and 6d). In contrast, we see very strong calibration on the Taiwanese real estate dataset (Figure 6g), although there are a few more outliers than included in the distribution.

We can conclude that the BNN manages the task of uncertainty prediction quite well. Hence, we recommend it to practitioners who are not satisfied with just a prediction, but also want a well-calibrated uncertainty estimation.

## F    TABLES OF MESSAGE EQUATIONS

In the following, we provide tables summarizing all message equations used throughout our model. The tables are divided into three categories: linear algebra operations (Table 5), training signals (Table 6), and activation functions (Table 7). Each table contains the relevant forward and backward message equations, along with illustrations of the corresponding factor graph where necessary. These summaries serve as a reference for the mathematical operations performed during inference and training, and they will be valuable for factor graph modeling across various domains beyond NNs.

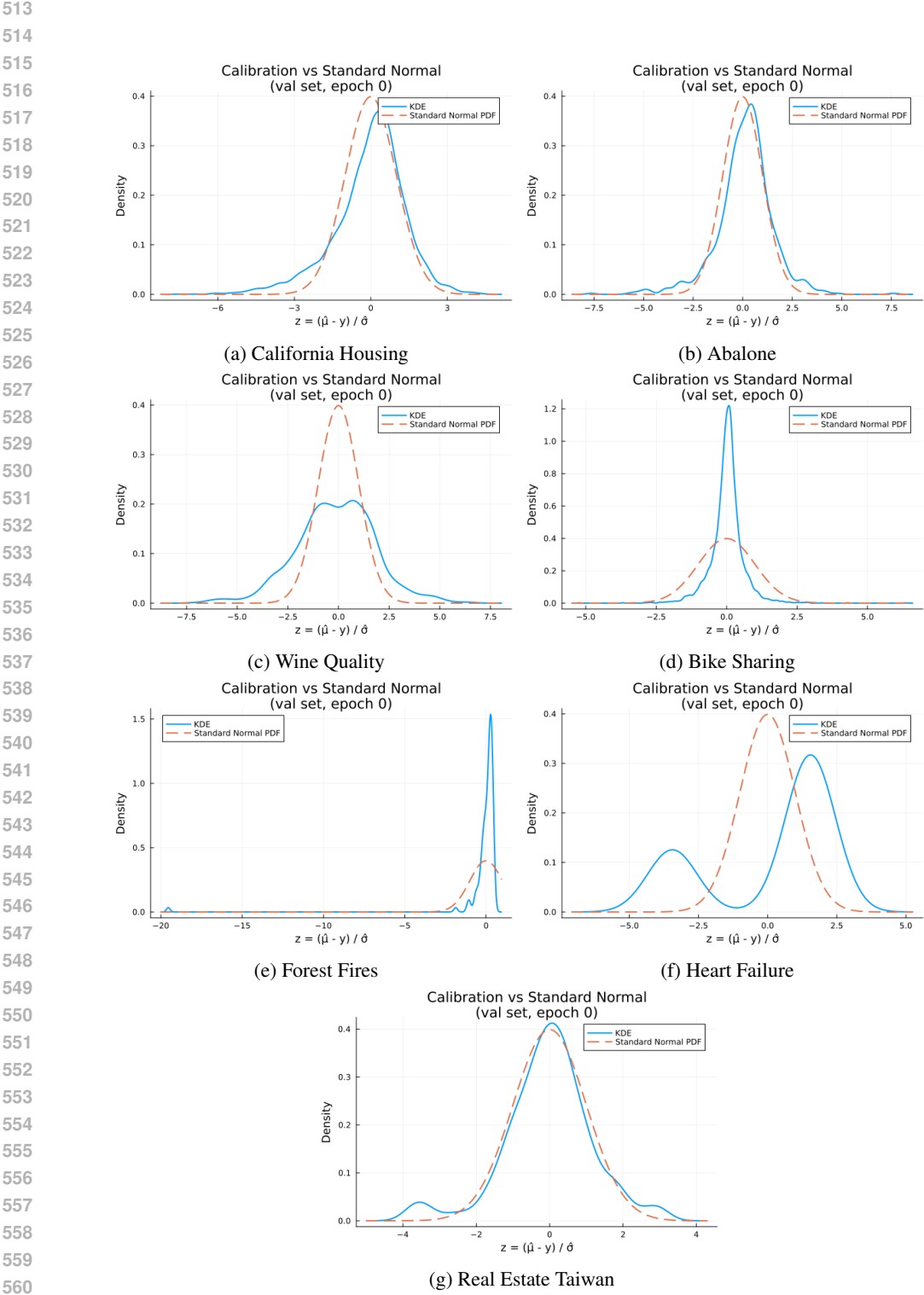

Figure 6: Calibration of the BNN: For each of the datasets, we plot the empiric z-score distribution and a standard normal distribution for reference. All plots are obtained for the validation dataset.

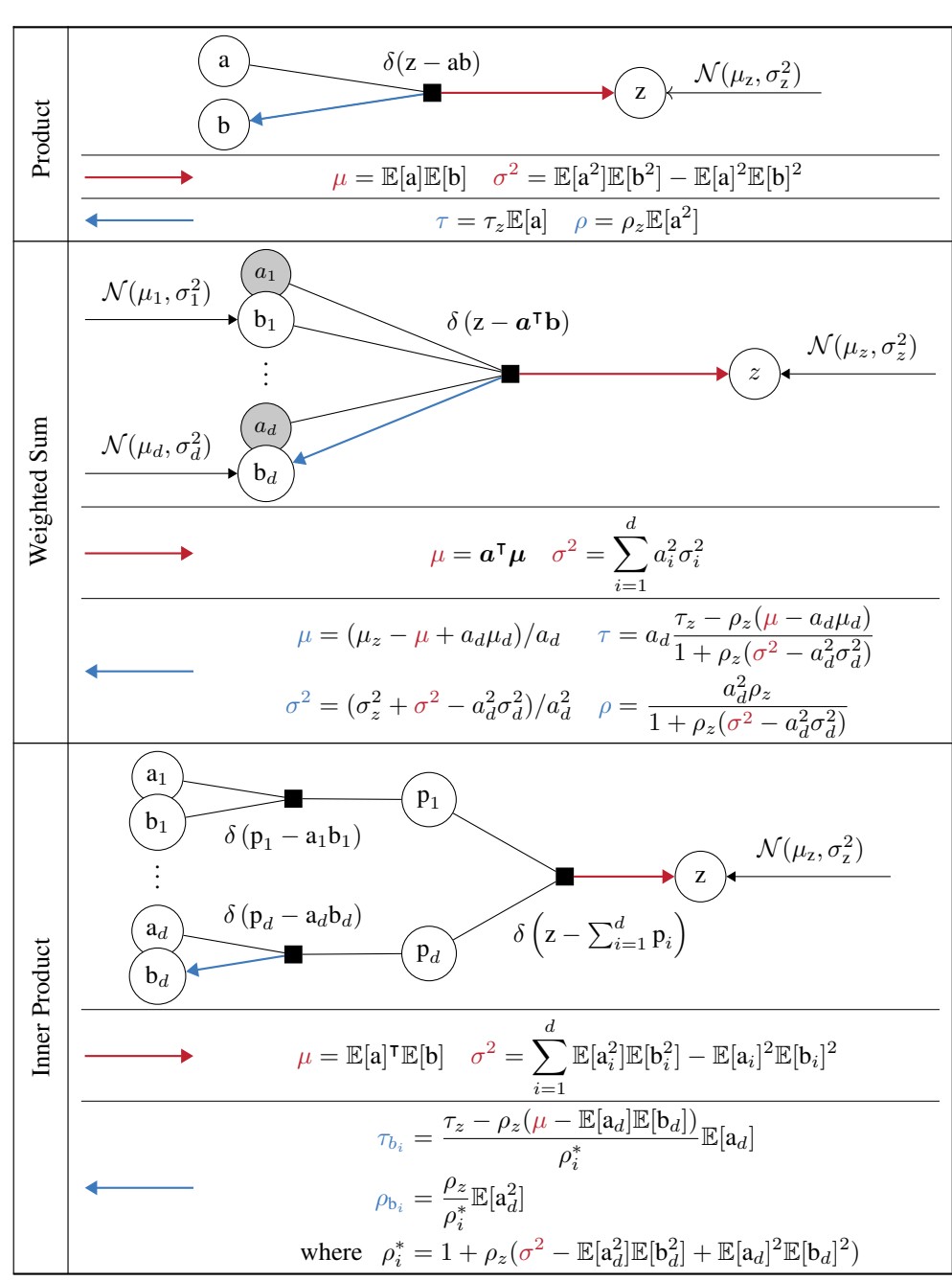

Table 5: Message equations for linear algebra: Calculating backward messages in natural parameters is preferable as it handles edge cases like $a_d = 0$ or $\rho_z = 0$ where location-scale equations are ill-defined. This approach also enhances numerical stability by avoiding division by very small quantities. Note that the inner product messages are simply compositions of the product and weighted sum messages with $a_i = 1$, $i = 1, \ldots, d$.

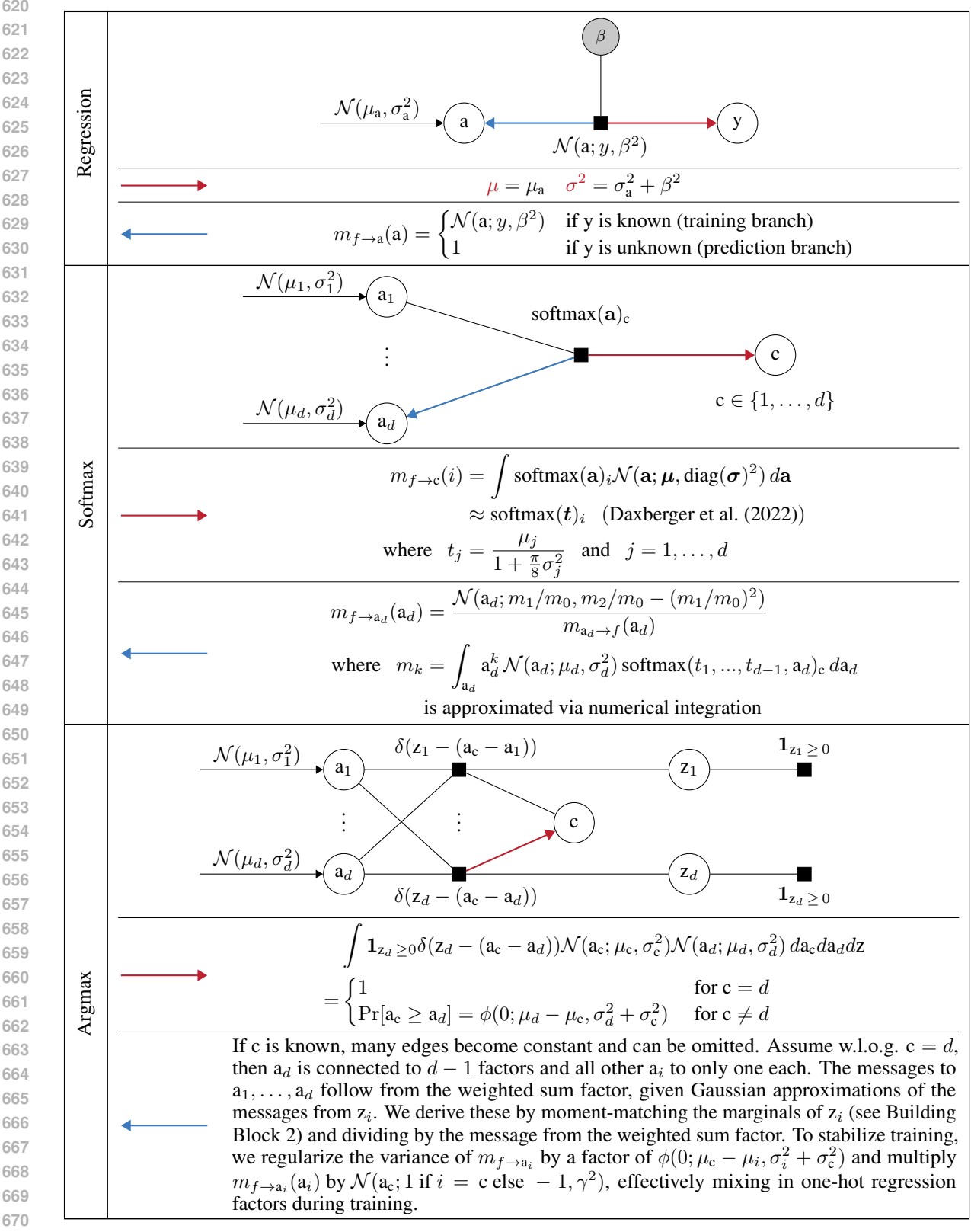

Table 6: Message equations for training signals. Note that the backward messages only apply in the case in which the target is known, i.e., on the training branches. On the prediction branch we only do foward passes.

| | |
|---|---|
| **Auxiliary Equations** | $$\text{ReLUMoment}_k(\mu, \sigma^2) = \begin{cases} \mathbb{E}[\text{ReLU(a)}] \text{ with a} \sim \mathcal{N}(\mu, \sigma^2) & \text{for } k = 1 \\ \mathbb{E}[\text{ReLU}^2(\text{a})] \text{ with a} \sim \mathcal{N}(\mu, \sigma^2) & \text{for } k = 2 \end{cases}$$ $$= \begin{cases} \sigma\varphi(x) + \mu\phi(x) & \text{for } k = 1 \\ \sigma\mu\varphi(x) + (\sigma^2 + \mu^2)\phi(x) & \text{for } k = 2 \end{cases}$$ where $\varphi$ and $\phi$ denote the pdf and cdf of $\mathcal{N}(0, 1)$, respectively. |
| | $$\zeta_k(\mu_1, \sigma_1, \mu_2, \sigma_2) := \int_0^\infty \text{a}^k \mathcal{N}(\text{a}; \mu_1, \sigma_1^2)\mathcal{N}(\text{a}; \mu_2, \sigma_2^2)\, d\text{a}$$ $$= \mathcal{N}(\mu_1; \mu_2, \sigma_1^2 + \sigma_2^2) \cdot \begin{cases} \text{ReLUMoment}_k(\mu_m, \sigma_m^2) & \text{for } k = 1, 2 \\ \phi(\mu_m/\sigma_m) & \text{for } k = 0 \end{cases}$$ with $\tau_m = \dfrac{\mu_1}{\sigma_1^2} + \dfrac{\mu_2}{\sigma_2^2}, \quad \rho_m = \dfrac{1}{\sigma_1^2} + \dfrac{1}{\sigma_2^2}, \quad \mu_m = \dfrac{\tau_m}{\rho_m}, \quad \text{and } \sigma^2 = \dfrac{1}{\rho_m}$ See Building Block 1 for the derivation of this equation. |
| **LeakyReLU** |  We use marginal approximation while: 1. The outputs are finite and not NaN 2. Forward message: Precision of $m_{f \to z}$ is $\geq$ precision of $m_{a \to f}$, and $m_0 > 10^{-8}$ 3. Backward message: It has worked well to require $(\tau_z > 0) \vee (\rho_z > 2 \cdot 10^{-8})$ Otherwise, we fall back to direct message approximation (forward) or $\mathbb{G}(0, 0)$ (backward). |
| **Direct** | $$\mu = (1 - \alpha) \cdot \text{ReLUMoment}_1(\mu_a, \sigma_a^2) + \alpha \cdot \mu_a$$ $$\sigma^2 = (1 - \alpha^2) \cdot \text{ReLUMoment}_2(\mu_a, \sigma_a^2) + \alpha^2 \cdot (\sigma_a^2 + \mu_a^2) - \mu^2.$$ |
| **Marginal** | $$m_{f \to z}(z) = \frac{\mathcal{N}(z; \frac{m_1}{m_0}, \frac{m_2}{m_0} - (\frac{m_1}{m_0})^2)}{m_{z \to f}(z)}$$ where $\quad m_k = (-1)^k \cdot \zeta_k(-\mu_a, \sigma_a^2, \ -\alpha \cdot \mu_z, \alpha^2 \cdot \sigma_z^2) + \zeta_k(\mu_a, \sigma_a^2, \ \mu_z, \sigma_z^2)$ To compute the marginal backward message, set $\alpha_{\text{back}} = \alpha^{-1}$ and swap $m_{a \to f}$ and $m_{z \to f}$ in the equation |

Table 7: Message equations for LeakyReLU with ReLU as the special case $\alpha = 0$.

