# OpenReview forum: "Approximate Message Passing for Bayesian Neural Networks"
_ICLR.cc/2026/Conference — Submitted to ICLR 2026_

### Official Review · Reviewer_kFYd · 2025-10-29

**Soundness:** 2
**Presentation:** 3
**Contribution:** 2
**Rating:** 4
**Confidence:** 4

**Summary:**

The paper proposes a novel Bayesian neural network (BNN) training framework based on **approximate message passing (MP)** over a factor graph representation of the predictive posterior. The key claim is that by modeling the network’s joint posterior via a factor graph with scalar latent variables and applying Gaussian message approximations, one can train BNNs (even convolutional nets) while accurately propagating uncertainty. Notably, the authors emphasize that their algorithm explicitly **avoids “double-counting”** of data as a known issue in prior MP-based BNN methods, which they argue has caused overconfident predictions in earlier approaches. The method is evaluated on several benchmarks. Empirically, the MP method achieves accuracy comparable to strong baselines while often exhibiting better calibration. The paper positions its contributions as: (1) a new MP framework with closed-form Gaussian message updates for all factors; (2) a practical implementation supporting CNNs and avoiding data double-counting; and (3) empirical evidence that this method is competitive with state-of-the-art approaches while improving uncertainty estimates.

**Strengths:**

1.	**Novel MP framework:** Introduces a principled factor-graph/BP approach for BNNs, deriving all necessary Gaussian message equations. This is a first step in applying belief propagation to large neural networks.

2.	**CNN support and data-debiasing:** Unlike earlier MP/EP methods, this framework handles convolutional layers and explicitly avoids “double-counting” of data during batch updates. This addresses a known cause of overconfident predictions in previous work.

3.	**Strong uncertainty calibration:** Empirically, the method yields excellent predictive uncertainty. For instance, on MNIST with limited data, MP achieved low expected calibration error (∼0.02) vs very high error for SGD. On CIFAR, MP’s ECE was the lowest among baselines. Synthetic tests show that credible intervals track true coverage well (correlation ~0.9). These demonstrate the method’s efficacy in capturing epistemic uncertainty.

4.	**Broad experimentation:** The authors evaluated on a variety of settings: tabular UCI regression, synthetic functions, MNIST and CIFAR image tasks. They compare to strong baselines (AdamW, IVON, deep ensembles) across multiple metrics (accuracy, NLL, Brier, OOD-AUC). This thoroughness lends credibility to their claims.

5.	**Clear writing of results:** The paper clearly reports results and highlights key findings (e.g. calibration advantage) in the text accompanying each table. The contributions and limitations are transparently listed.

**Weaknesses:**

1.	**Heaviness of approximations:** The core MP algorithm uses Gaussian approximations for all messages without theoretical guarantees. Such approximations in highly nonlinear, loopy graphs may introduce bias. The paper does not analyze the accuracy or failure modes of these approximations, leaving uncertainty about when MP might break down.

2.	**Scalability concerns:** The method is computationally expensive. As acknowledged by the authors, training is up to two orders of magnitude slower and more memory-intensive than standard SGD/AdamW. Each training example carries a full set of messages, and the Julia implementation uses double-precision for safety, making it impractical for very large models.

3.	**Limited experimental breadth:** The neural architecture used is relatively small (890K parameters, plain 6-layer CNN). Modern benchmarks (ResNet, Transformers) with residuals or normalization are not evaluated. It is unclear how the method handles practical architectures (batch-norm, skip-connections). This restricts claims of general applicability.

4.	**Mixed empirical performance:** Although calibration is good, the MP method’s predictive accuracy and NLL are inferior to top alternatives. For example, on CIFAR-10 Deep Ensembles achieve 81.9% accuracy vs 77.3% for MP. The paper notes that differences in architecture (lack of normalization/residuals) may explain this, but it underscores that MP does not yet match state-of-art predictive performance in high-dimensional tasks.

5.	**Reproducibility details:** The paper defers many architectural and hyperparameter details to the Appendix. Important choices (e.g. noise priors, initialization variances) are only briefly mentioned. The lack of multiple runs or error bars also raises questions about robustness.

**Questions:**

1.	**Double-counting fix:** Can you clarify the theoretical justification for the batch-update rule (dividing out old aggregate messages) that prevents double-counting? Does this exactly recover Bayesian posterior updates, or is it a heuristic? How sensitive is training to this scheme?

2.	**Scalability to modern nets:** Have you tried incorporating residual connections or normalization (e.g. BatchNorm)? You mention a possible factor for division, but has this been validated? Can the MP algorithm scale to architectures like ResNets or Transformers?

3.	**Multiple passes or convergence:** How many MP iterations (sweeps through the factor graph) are performed per batch? Is the algorithm guaranteed to converge, or do you observe oscillations? Have you considered damping or other techniques to stabilize message updates?

4.	**Baselines and hyperparameters:** How were AdamW and IVON tuned? Are the baselines given the best possible schedules and hyperparameters? For CIFAR, why only 25 epochs? Would longer training change the comparisons?

5.	**Prediction procedure:** For classification, you mention an “argmax factor” in Appendix F. Does this mean you approximate the softmax? How are predictive probabilities computed at test time? Clarify how classification uncertainty is obtained from the Gaussian-approximated network outputs.

6.	**Posterior quality diagnostics:** Beyond ECE and credible intervals, did you assess other uncertainty metrics (e.g. negative log-likelihood on OOD, Brier decomposition)? Did you examine any divergences or posterior samples to verify that the approximate posterior makes sense (e.g. covariance structure)?

**Details Of Ethics Concerns:**

No ethical issues were identified.

---

### Official Review · Reviewer_JWMe · 2025-10-30

**Soundness:** 1
**Presentation:** 1
**Contribution:** 2
**Rating:** 2
**Confidence:** 4

**Summary:**

The paper introduces a novel approximate message-passing framework for Bayesian Neural Network inference that scales to architectures such as CNNs and MLPs. Unlike earlier approaches such as Expectation Backpropagation (EBP) and Probabilistic Backpropagation (PBP), the proposed method explicitly avoids double-counting of training data and prevents posterior collapse. The framework models BNNs as factor graphs and derives Gaussian message approximations for all relevant factors. The framework is tested on MNIST, CIFAR-10, and  UCI regression benchmarks, comparing to baselines like SGD, IVON, and Deep Ensembles.

**Strengths:**

1. The work attempts to address a core flaw in prior message-passing BNNs (double-counting), which may lead to posterior collapsing

2. The work extends message passing to CNNs, while earlier works were limited to MLPs.

3. The framework formulates BNN inference as belief propagation in a factor graph with analytically tractable Gaussian message approximations, which is computationally efficient.

**Weaknesses:**

1. I unfortunately find that the empirical validation is insufficient to fully support the paper’s claims.
    * The authors claim their message-passing framework alleviates overconfidence and posterior collapse compared to prior approaches such as EBP and PBP. However, these methods are not included as experimental baselines, making it difficult to verify those claims empirically.
    * The choice and coverage of baselines are inconsistent across experiments. For example, Figure 3a omits R-SGD, Figure 3b shows Deep Ensemble results only, and Figures 3c, 3d exclude AM-MP and R-SGD. It is also unclear which architecture (MLP or LeNet-5) Figure 3 corresponds to.
    * In Table 4, the use of RMSE alone is insufficient to assess calibration quality or uncertainty estimation. Complementary metrics such as coverage or NLL would strengthen the evidence.

2. There might be potential fragility and computational cost due to numerous numerical stabilizations. The framework relies on several ad-hoc stability guardrails (Section 4), including special handling for activations like LeakyReLU, re-normalization of small constants, and periodic recomputation of weight marginals. While these measures improve numerical stability, they may introduce additional computational overhead and raise concerns about the robustness and generalizability of the approach across architectures or datasets.

 3. Minor on the writing side. There are a lot of acronyms that are used without being introduced or introduced later, such as IVON, AM-MP

**Questions:**

1. How are regression-based MPs trained on classification tasks?
2. Can the "double-counting" issue of EBP and PBP be overcome by using the tempered posterior [1]



[1] Ng, Kenyon, et al. "Temperature Optimization for Bayesian Deep Learning." arXiv preprint arXiv:2410.05757 (2024).

---

### Official Review · Reviewer_bdZB · 2025-10-31

**Soundness:** 3
**Presentation:** 2
**Contribution:** 2
**Rating:** 4
**Confidence:** 4

**Summary:**

This paper proposes a message-passing framework for Bayesian Neural Networks (BNNs) that reformulates training as inference on a factor graph using a diagonal-Gaussian approximation family. It addresses the double-counting problem in existing variational inference methods by introducing a batch-wise “divide-then-multiply” update scheme and a layer-level implementation. The authors implement the method in Julia with GPU support, demonstrating its scalability to MLPs and CNNs and improved calibration over AdamW, IVON, and Deep Ensembles on small-scale benchmarks.

**Strengths:**

1. The paper introduces a message-passing framework for Bayesian neural networks that provides a principled way to model uncertainty while mitigating issues such as data double-counting and overconfidence observed in earlier Bayesian inference methods.

2. The framework is mathematically coherent, deriving closed-form Gaussian message updates for common factor types and supporting deterministic training without reliance on sampling-based approximations.

3. Experiments demonstrate that the proposed approach produces well-calibrated uncertainty estimates and competitive predictive performance across several benchmarks.

**Weaknesses:**

1. Expanding one convolutional kernel into tens of thousands of scalar $\delta$ factors substantially increases the size of the factor graph so that memory usage is tied to the count of scalar edges, not to the parameter count. This raises practical scalability problems for deep or wide CNNs.

2. Each ReLU layer is handled by moment-matched Gaussian messages. But since the exact moments are computed under a Gaussian assumption that ignores the truncated tail, the resulting estimates may introduce small systematic biases. These biases could accumulate over multiple layers, potentially affecting calibration accuracy in deeper models.

3. The batch-wise “divide-then-multiply” update operates in precision space by dividing the current marginal by the old batch message. When the old message carries low precision, this division amplifies rounding errors and often produces negative precisions that must be clamped to preserve numerical validity. These corrective heuristics reveal intrinsic instability in the update rule, which could undermine convergence and consistency across training iterations.

**Questions:**

1. Have you tested any sparse or tensor-factorised representations that keep a single factor per convolutional kernel element instead of one $\delta$ -node per scalar multiply–add, and if so what reduction in peak GPU memory did you observe without changing the converged marginal means?

2. Have you checked the skewness or other higher-order moments of the ReLU messages to verify that the Gaussian approximation remains adequate across depth?

---

### Official Review · Reviewer_mpkr · 2025-10-31

**Soundness:** 2
**Presentation:** 2
**Contribution:** 2
**Rating:** 4
**Confidence:** 4

**Summary:**

This paper trains BNNs using “approximate message passing on factor graphs”: it combines Gaussian messages in batches to avoid duplicate counting, scales to CNNs, and improves calibration and OoD performance, but training and memory costs remain high

**Strengths:**

1. Laplace/EP-style Gaussian messages makes stable training and avoid posterior collapse
2. Batch message aggregation mitigates double-counting and better calibration than AdamW/ensembles

**Weaknesses:**

1. Computational and memory overhead is high, as Gaussian messages are maintained per sample, resulting in scalability that falls short of standard training pipelines like SGD/Adam
2. Insufficient large-scale validation, lacking systematic comparisons and ablation studies on larger datasets/modern benchmarks
3. The processing of residuals, normalization, and other modern layers remains unclear, and there is insufficient evidence of generalization beyond simple CNNs
4. Loopy BP lacks convergence guarantees. Although the authors acknowledge that guarantees only hold under specific conditions (such as Simon's condition) and are difficult to verify, still remains some concerns

**Questions:**

The computational power/graphics memory cost of the method is excessively high, its scalability evidence is insufficient, and its effectiveness and stability in medium-to-large-scale, modern networks (residual/normalized) and larger datasets remain poorly substantiated by robust empirical evidence

---

### Meta-Review · Area_Chair_x8VD · 2026-01-06

**Summary:**

This paper proposes a message-passing framework to train Bayesian neural networks. Reviewers expressed concerns about computational cost and empirical validation. The authors did not provide a rebuttal, and I thus recommend rejection.

**Reviewer Concerns:**

Reviewer concerns were not addressed.

**Reviewer Scores:**

Reviewers would have kept their scores.

---

### Decision · Program_Chairs · 2026-01-26

Reject